# Inverting Data Transformations via Diffusion Sampling

**Jinwoo Kim** [* 1]  **Sékou-Oumar Kaba** [* 2 3]  **Jiyun Park** [1]  **Seunghoon Hong** [† 1]  **Siamak Ravanbakhsh** [† 2 3]

## Abstract

We study the problem of *transformation inversion* on general Lie groups: a datum is transformed by an unknown group element, and the goal is to recover an inverse transformation that maps it back to the original data distribution. Such unknown transformations arise widely in machine learning and scientific modeling, where they can significantly distort observations. We take a probabilistic view and model the posterior over transformations as a Boltzmann distribution defined by an energy function on the data space. To sample from this posterior, we introduce a diffusion process on Lie groups that keeps all updates on-manifold and only requires computations in the associated Lie algebra. Our method, *Transformation-Inverting Energy Diffusion* (TIED), relies on a new trivialized target-score identity that enables efficient score-based sampling of the transformation posterior. As a key application, we focus on *test-time equivariance*, where the objective is to improve the robustness of pretrained neural networks to input transformations. Experiments on image homographies and PDE symmetries demonstrate that TIED can restore transformed inputs to the training distribution at test time, showing improved performance over strong canonicalization and sampling baselines.

## 1. Introduction

Many data modalities are observed after unknown or unusual geometric transformations. We ask a general question: given data transformed by an unknown element of a Lie group, can we recover an inverse transformation based on a given data distribution? This problem appears in many scenarios, including viewpoint changes and projective warps in computer vision, sensor motion and registration in medical imaging, and changes of reference frame in scientific modeling (Olver, 1993; Hartley & Zisserman, 2003; Celledoni et al., 2021). It is an instance of a blind inverse problem since the nuisance transform is unknown and does not admit a unique solution in general. Probabilistic methods are therefore a natural fit (Kaipio & Somersalo, 2006). Unlike existing approaches for blind inverse problems that sample in data space (Chung et al., 2022) or are specialized to particular transformation families (Zitova & Flusser, 2003; Beg et al., 2005; Hartley et al., 2013), we tackle the problem for general Lie group actions, with all inference and sampling done on the group.

We model inverse transformations using energy-based models that quantify the likelihood of different transformations of the data. This formulation allows us to leverage pretrained energy-based models or energies designed using domain expertise. Sampling from energy-based models can however be slow and challenging, especially when the energy landscape is rugged and multimodal. To address this, we introduce a new sampling method on Lie groups based on diffusion (Song et al., 2020), which follows scores along a noise schedule. We show that the sampling can be performed on the Lie group by estimating scores in the Lie algebra (Fig. 1), extending trivialization approaches for handling curved manifolds (Lezcano Casado, 2019; Zhu et al., 2025). Our method covers a very general class of groups without assuming compactness, a bi-invariant metric, or linear actions.

As a key application, we focus on the case where transformed data are inputs to a neural network. It is known that pretrained networks can exhibit a marked lack of robustness to geometric transformations (Hendrycks & Dietterich, 2019; Ollikka et al., 2024). Equivariant neural networks address this for some groups and modalities (Cohen & Welling, 2016; Ravanbakhsh et al., 2017; Kondor & Trivedi, 2018; Finzi et al., 2021), but require highly specialized architectures that can be difficult to scale. Instead of using equivariant models, we wish to improve the robustness of generic, pretrained networks. The ability to invert data transformations ensure that we can bring samples back in-distribution and that the model does not perform inference on out-of-distribution transformed data. Prior methods based on deterministic canonicalization (Jaderberg et al., 2015; Esteves et al., 2017; Kaba et al., 2023) pursue this idea but either rely on training additional equivariant networks (Mondal et al.,

---

*Equal contribution † Equal advising [1]KAIST, South Korea
[2]Mila - Quebec Artificial Intelligence Institute, Montréal, Canada
[3]School of Computer Science, McGill University, Montréal, Canada.
Correspondence to: Jinwoo Kim <jinwoo-kim@kaist.ac.kr>, Sékou-Oumar Kaba <sekou.oumar.kaba@gmail.com>.

*Proceedings of the $43^{rd}$ International Conference on Machine Learning*, Seoul, South Korea. PMLR 306, 2026. Copyright 2026 by the author(s).

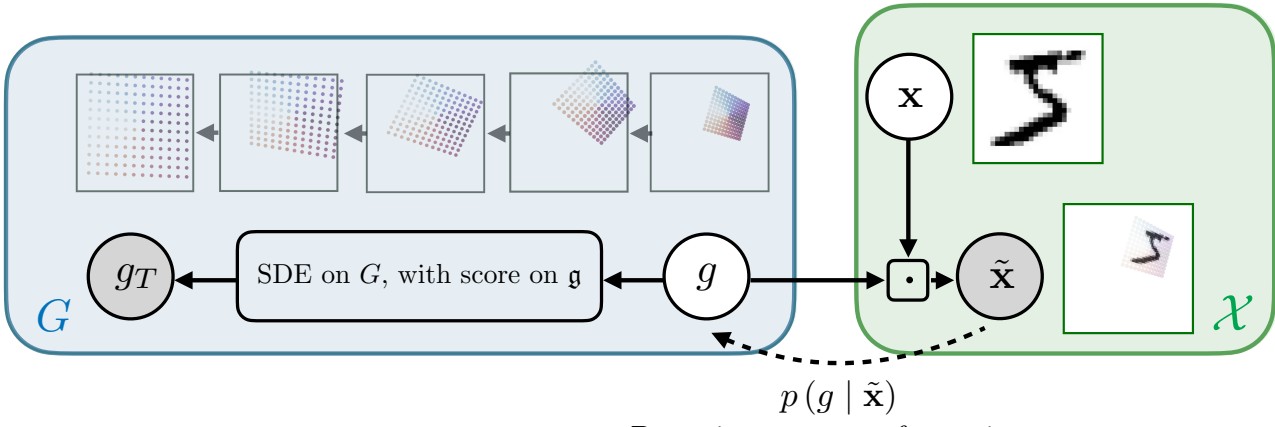

*Figure 1.* Graphical model describing our problem and method (with observed variables in gray and unobserved variables in white). $\mathcal{X}$ denotes the data space and $G$ a group of transformations, here the group of image homographies $\mathrm{PGL}(3, \mathbb{R})$. We are provided a data sample $\tilde{\mathbf{x}}$ that is generated by transforming an unknown in-distribution sample $\mathbf{x}$ with an unknown transformation $g$. We wish to sample from the posterior over transformations. Inspired by diffusion models, we construct a fast sampler that reverts a diffusion process on the Lie group starting from a random group element. The scores of the SDE are computed in the Lie algebra $\mathfrak{g}$.

2023) or use optimization methods (Schmidt & Stober, 2024; Shumaylov et al., 2024; Singhal et al., 2025) which can be brittle and do not readily extend to general Lie groups.

Our method allows us to make any pretrained model equivariant to Lie groups at test-time using an energy-based model. The energy can be approximated via the pretrained model itself (Grathwohl et al., 2020), which results in a training-free pipeline. This can be interpreted as a form of inference-time scaling for equivariance. The inference-time cost comes from sampling in-distribution transformations, which is made efficient and fast with our novel diffusion sampler.

Our **main contributions** are as follows[1]:

1. We introduce Transformation-Inverting Energy Diffusion (TIED), a novel diffusion sampler for inverting data transformations. It is guaranteed to stay on-manifold (i.e., exactly transform the input data) and handles general Lie groups and nonlinear actions, only requiring knowledge of the Lie algebra.

2. We prove that inversion of data transformations can be used for test-time equivariance of pretrained models, only requiring an approximate energy model.

3. We demonstrate that our method can invert transformations on challenging groups such as image homographies and Lie point symmetries for partial differential equations (PDEs). On these problems, we demonstrate that our method consistently outperforms baselines in improving the performance of pretrained networks.

---

[1]Code is available at https://github.com/jw9730/tied.

## 2. Background and Related Work

### 2.1. Inverse problems on groups

Recovering unknown transformations has a long history in problems that are naturally formulated on a group. In alignment and rotation synchronization, the aim is to recover poses on orthogonal or Euclidean groups from pairs of relative measurements (Singer, 2011; Hartley et al., 2013). Related are also image registration problems in which one seeks to recover the affine or homography transformation between pairs of images (Zitova & Flusser, 2003; Beg et al., 2005; Lorenzi & Pennec, 2013). In contrast to our setting, in these problems we are often provided pairs of samples, as opposed to a single one. In addition, these methods typically rely on group-specific knowledge. A recent line of work proposes to use generative models to recover inverse samples (Venkatakrishnan et al., 2013; Kadkhodaie & Simoncelli, 2020; Song et al., 2020; Chung et al., 2022). However, since they operate in data space instead of the group, they require large diffusion models and, for non-linear actions, do not recover transformations that lie on the manifold.

### 2.2. Test-time robustness and canonicalization

Canonicalization methods improve robustness of neural networks by mapping inputs to a reference pose before prediction (Jaderberg et al., 2015; Esteves et al., 2017; Kaba et al., 2023; Mondal et al., 2023). These methods offer an alternative paradigm to equivariant neural networks (Cohen & Welling, 2016; Bronstein et al., 2021) and data augmentation (Shorten & Khoshgoftaar, 2019; Cubuk et al., 2019) that does not require specialized architectures or more expensive training procedures. Canonicalization yields equivariance

for any predictor if the canonicalizer is itself equivariant. Previous works have proposed probabilistic variants (Kim et al., 2023; Dym et al., 2024; Cornish, 2024; Lawrence et al., 2025), but they rely on equivariant networks and are therefore limited in applications. Most closely related to our method are optimization-based canonicalization methods (Kaba et al., 2023; Schmidt & Stober, 2024; Shumaylov et al., 2024; Singhal et al., 2025), which obtain an equivariant canonicalizer without requiring equivariant primitives. They are, however, purely deterministic, often specialize in specific groups and necessitate optimization of highly rugged energy functions. Test-time data augmentation methods are an alternative (Krizhevsky et al., 2012; Szegedy et al., 2015; Wang et al., 2019; Kim et al., 2020; Shanmugam et al., 2021), and improve prediction through an ensembling effect (Hansen & Salamon, 2002). They, however, lack equivariance guarantees and use heuristic augmentation distributions. Group averaging (Yarotsky, 2022) and its frame-based variant (Puny et al., 2021) guarantee equivariance, yet are often intractable for Lie groups and may average over out-of-distribution transformations.

### 2.3. Diffusion sampling

Diffusion models (Sohl-Dickstein et al., 2015; Song et al., 2020) are a class of generative models that produce samples by first sampling from a simple base distribution $p_1(\mathbf{x})$ and integrating along the stochastic differential equation (SDE)

$$\mathrm{d}\mathbf{x}_t = -\gamma(t)^2 \, \nabla_{\mathbf{x}_t} \log p_t(\mathbf{x}_t) \, \mathrm{d}t + \gamma(t) \, \mathrm{d}\bar{\mathbf{w}}_t,$$

where $t \in [0,1]$, $\gamma(t)$ is a scalar diffusion coefficient, and $\bar{\mathbf{w}}_t$ is Brownian motion run backward in time. $p_t(\mathbf{x}_t)$ is a marginal 'noisy' distribution under the forward process

$$\mathrm{d}\mathbf{x}_t = \gamma(t) \, \mathrm{d}\mathbf{w}_t, \tag{1}$$

starting from the target distribution $p_0(\mathbf{x}) \equiv p(\mathbf{x})$. We consider here the variance-exploding (VE) SDE. In generative modeling, the scores $\nabla_{\mathbf{x}_t} \log p_t(\mathbf{x}_t)$ can be estimated from data (Vincent, 2011). Since the scores of the noisy densities are smoother than those of the original density and non-vanishing in regions of low energy, diffusion-based sampling has an advantage over more traditional score-based MCMC (Song & Ermon, 2019). The success of diffusion models has inspired a variety of methods for *sampling* from Boltzmann distributions using reverse SDEs; see, e.g., Richter & Berner (2023); Vargas et al. (2023); Akhound-Sadegh et al. (2024); De Bortoli et al. (2024). These methods enable the estimation of scores along the probability path, assuming access to the energy and its gradients, rather than training data samples. We discuss later extensions to Lie groups, which are relevant to our problem. Diffusion models over transformations have also been proposed (Bansal et al., 2023), but unlike our method, use deterministic non-invertible transformations.

## 3. Probabilistic Transformation Inversion

### 3.1. Preliminaries

**Lie groups** We consider a set of transformations $G$ forming a connected Lie group. While a Lie group is a curved manifold, a lot of its properties derive from its tangent space at the identity, the Lie algebra, denoted by $\mathfrak{g} \equiv T_e G$. We denote by $\exp : \mathfrak{g} \to G$ the exponential map and by $\mu$ the Haar measure on $G$. The left multiplication map $L_g : G \to G$ is defined as $L_g : h \mapsto gh$, and its tangent map $\mathrm{d}(L_g)_h : T_h G \to T_{gh} G$ is one-to-one. We highlight two cases:

$$\mathrm{d}(L_g)_e : \mathfrak{g} \to T_g G, \quad \mathrm{d}(L_{g^{-1}})_g : T_g G \to \mathfrak{g}.$$

These tangent maps are useful for working in the Lie algebra instead of the tangent spaces of arbitrary group elements, a technique called *(left-)trivialization* in the literature (Lezcano Casado, 2019; Kong & Tao, 2024). A particular example we frequently use is the trivialized gradient operator $\nabla_{\mathfrak{g}} \equiv \mathrm{d}(L_{g^{-1}})_g \nabla$, which takes a function on $G$ and evaluates its gradient in $\mathfrak{g}$. Notably, it can be computed with standard automatic differentiation tools without the need to explicitly handle tangent maps. We refer the reader to §A for additional background on Lie groups and trivialization.

**Group actions** We consider data samples $\mathbf{x} \in \mathcal{X}$ and outputs $\mathbf{y} \in \mathcal{Y}$, where $\mathcal{X}, \mathcal{Y} \subseteq \mathbb{R}^d$. The data density is denoted $p_{\mathbf{x}}(\mathbf{x})$. We assume a diffeomorphic but not necessarily linear action of $g \in G$ on data samples $\phi_g(\mathbf{x})$ also denoted by $g \cdot \mathbf{x}$. The Jacobian of the group action $\phi_g$ evaluated at $\mathbf{x}$ is denoted by $J_g(\mathbf{x})$. For linear actions, $J_g$ is a group representation independent of $\mathbf{x}$. The orbit of $\mathbf{x}$ is the set of samples that can be obtained through transformations and is denoted $G \cdot \mathbf{x}$. We denote by $\mu_{G \cdot \mathbf{x}}$ the pushforward of the Haar measure using $\phi_g(\mathbf{x})$. A function $f : \mathcal{X} \to \mathcal{Y}$ is equivariant if $f(g \cdot \mathbf{x}) = g \cdot f(\mathbf{x})$ for all $g \in G, \mathbf{x} \in \mathcal{X}$ and invariant if $f(g \cdot \mathbf{x}) = f(\mathbf{x})$. Similarly, a conditional density is equivariant if $p(g \cdot \mathbf{y} \mid g \cdot \mathbf{x}) = p(\mathbf{y} \mid \mathbf{x}) |\det J_g(\mathbf{y})|^{-1}$ for all $g \in G, \mathbf{x} \in \mathcal{X}, \mathbf{y} \in \mathcal{Y}$.

### 3.2. Problem statement

We now formally introduce the problem of transformation inversion. We assume that we are given an out-of-distribution sample $\tilde{\mathbf{x}} = g \cdot \mathbf{x}$, generated by applying an unknown transformation $g$ to an in-distribution sample $\mathbf{x}$. Our aim is to solve the blind inverse problem and recover the pair $(g, \mathbf{x})$. Since this problem does not admit a unique solution, a probabilistic approach is a natural choice. Assuming a uniform prior over the unknown transformation, a classical solution to inverse problems (see e.g., Stuart, 2010) is to consider the Bayesian posterior,

$$p(\mathbf{x} \mid \tilde{\mathbf{x}}) = \frac{p_{\mathbf{x}}(\mathbf{x}) \, p(\tilde{\mathbf{x}} \mid \mathbf{x})}{p(\tilde{\mathbf{x}})} \propto p_{\mathbf{x}}(\mathbf{x}) \, \mathbb{1}_{G \cdot \mathbf{x}}(\tilde{\mathbf{x}}),$$

where the density is with respect to $\mu_{G \cdot \mathbf{x}}$ and $\mathbb{1}_{G \cdot \mathbf{x}}$ is the indicator function over the orbit of $\mathbf{x}$.

Rather than modeling the posterior in data space, we can alternatively model the posterior $p(g \mid \tilde{\mathbf{x}})$ directly on the group. This has the advantage that the group is (often significantly) lower dimensional than the data space, and that this allows recovering the inverse transformation $g$. We can show that the posterior is given by the following.

**Proposition 3.1** (Transformation inversion posterior). *Let $G$ act freely on $\mathcal{X}$. Then,*

1. *The posterior distribution of $g$ has density $p(g \mid \tilde{\mathbf{x}}) \propto p_{\mathbf{x}}\left(g^{-1} \cdot \tilde{\mathbf{x}}\right) |\det J_{g^{-1}}(\tilde{\mathbf{x}})|$.*

2. *The random variable $\mathbf{x}' = h^{-1} \cdot \tilde{\mathbf{x}}$, with $h \sim p(g \mid \tilde{\mathbf{x}})$ has density $p(\mathbf{x}' \mid \tilde{\mathbf{x}}) \propto p_{\mathbf{x}}(\mathbf{x}') \mathbb{1}_{G \cdot \tilde{\mathbf{x}}}(\mathbf{x}')$ w.r.t. $\mu_{G \cdot \tilde{\mathbf{x}}}$.*

A proof is in §B.1. For the first result, it expresses $p(\tilde{\mathbf{x}} \mid g)$ with $p_{\mathbf{x}}$ and a Jacobian correction and applies Bayes' rule with a uniform prior, and for the second, it makes a change of variables that cancels out the Jacobian. This result shows that when a prior model of the data distribution $p_{\mathbf{x}}$ is available, the posterior over transformations can be directly recovered. Additionally, sampling $h \sim p(g \mid \tilde{\mathbf{x}})$ and canonicalizing the out-of-distribution sample via $h^{-1} \cdot \tilde{\mathbf{x}}$ yields samples from the prior data distribution, which conforms to intuition. The free action assumption simplifies the proof, and does not impose a limitation in practice, as we discuss in §B.1.

Writing the posterior over $g$ as a Boltzmann density, we have

$$p(g \mid \tilde{\mathbf{x}}) = e^{-E_{\mathbf{x}}\left(g^{-1} \cdot \tilde{\mathbf{x}}\right)} |\det J_{g^{-1}}(\tilde{\mathbf{x}})| / Z(\tilde{\mathbf{x}}),$$
$$Z(\tilde{\mathbf{x}}) = \int_G e^{-E_{\mathbf{x}}\left(g^{-1} \cdot \tilde{\mathbf{x}}\right)} |\det J_{g^{-1}}(\tilde{\mathbf{x}})| \mathrm{d}\mu(g), \quad (2)$$

where $E_{\mathbf{x}}(\mathbf{x}) = -\log p_{\mathbf{x}}(\mathbf{x})$ is the energy associated with the data prior. Under mild conditions on the energy, the normalization constant $Z(\tilde{\mathbf{x}})$ is finite even for non-compact groups and the density is well-defined. Note that any function of $\tilde{\mathbf{x}}$ can be added to the energy without changing the density, since it will cancel in the normalization. Therefore, the energy does not need to be accurate across orbits, but only within each orbit. This is significantly simpler than modeling the data space and is closer to a generative model over transformations (Allingham et al., 2024).

### 3.3. Application to test-time equivariance

We now consider an application of our framework in improving the robustness of a pretrained neural network $f_\theta : \mathcal{X} \to \mathcal{Y}$. A common failure mode of neural networks, including those trained on large datasets and with data augmentation, is their poor generalization to transformed samples (Hendrycks & Dietterich, 2019). By investing a small amount of compute

at test time to sample an inverse transformation, we aim to improve the pretrained model.

Following the canonicalization method (Kaba et al., 2023), we define the test-time predictor $\tilde{f}$ as

$$\tilde{f}(\tilde{\mathbf{x}}) = g \cdot f\left(g^{-1} \cdot \tilde{\mathbf{x}}\right), \quad g \sim p(g \mid \tilde{\mathbf{x}}),$$

where the input to the pretrained model $g^{-1} \cdot \tilde{\mathbf{x}}$ is guaranteed to lie in distribution by Prop. 3.1. Results by Bloem-Reddy & Teh (2020); Lawrence et al. (2025) show that this randomized predictor is equivariant if $p(g \mid \tilde{\mathbf{x}})$ is an equivariant conditional distribution. The natural question is therefore, does the transformation inversion posterior satisfy conditional equivariance? This is indeed the case, which ensures the soundness of our approach.

**Proposition 3.2** (Equivariance of the posterior). *For any $E_{\mathbf{x}} : \mathcal{X} \to \mathbb{R}$, the posterior density $p(g \mid \tilde{\mathbf{x}})$ is $G$-equivariant.*

A proof is in §B.2, showing that if the sample $\tilde{\mathbf{x}}$ is transformed, then the posterior must change in the opposite way to ensure that $g^{-1} \cdot \tilde{\mathbf{x}}$ is still distributed according to the data prior. An intuitive explanation is that transforming $\tilde{\mathbf{x}} \mapsto h \cdot \tilde{\mathbf{x}}$ and $g \mapsto h \cdot g$ leaves the data energy $E_{\mathbf{x}}(g^{-1} \cdot \mathbf{x})$ unchanged, as the transformation cancels out.

Following Kim et al. (2023), we can also sample an ensemble of diverse in-distribution samples and average predictions over them when $p(g \mid \tilde{\mathbf{x}})$ is equivariant:

$$\tilde{f}(\tilde{\mathbf{x}}) = \mathbb{E}_{g \sim p(g \mid \tilde{\mathbf{x}})}\left[g \cdot f\left(g^{-1} \cdot \tilde{\mathbf{x}}\right)\right]. \quad (3)$$

It has been shown that ensembling over augmentations can further improve the performance of pretrained neural networks (Shanmugam et al., 2021). Our method provides a principled form for the augmentation distribution $p(g \mid \tilde{\mathbf{x}})$, whereas most methods use distributions motivated by heuristics. In practice, we use an ensembling strategy that selects the lowest-energy augmentation, which can be interpreted as sampling from a sharpened distribution then doing one-sample estimation of $\tilde{f}$. We discuss this in detail in §B.10.

There is significant flexibility in the choice of the energy function $E_{\mathbf{x}}$. Following optimization-based canonicalization methods (Schmidt & Stober, 2024; Singhal et al., 2025), we consider defining the energy directly from the pretrained model $f_\theta$ using its prediction confidence if it is a classifier (Grathwohl et al., 2020). We also consider the evidence lower bound (ELBO) approximation of the energy using variational autoencoders (VAEs) (Kingma & Welling, 2013; Shumaylov et al., 2024).

## 4. Transformation-Inverting Energy Diffusion

### 4.1. Challenges in sampling from general Lie groups

The next question is how to sample from the transformation-inversion distribution (2), assuming we have access to an

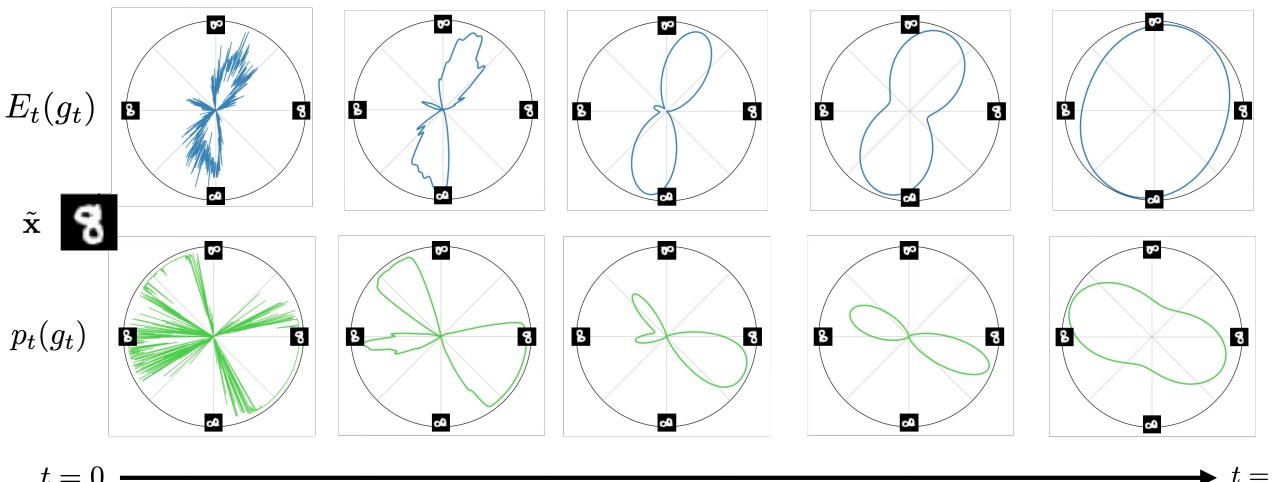

*Figure 2.* Energy (top) and density (bottom) along the forward process (5) for the group of rotations $G = \mathrm{SO}(2)$. The energy of the prior $E_0(g) \equiv E_{\mathbf{x}}\left(g^{-1} \cdot \tilde{\mathbf{x}}\right)$ is defined using the `logsumexp` of classifier logits from a ResNet18 trained on MNIST. The energy at small timesteps (top left) is low for likely orientations of the MNIST digit $\tilde{\mathbf{x}}$. The posterior $p_0(g)$ (bottom left) has modes centered around the most likely transformations. Since the energy is obtained from a neural network, its landscape is highly rugged, resulting in exploding and vanishing scores. Going to the right, the densities along the forward process are plotted. The landscapes are increasingly smooth and address the ruggedness.

energy-based model and can take its gradient. This is challenging for two reasons. First, the energy landscape can be highly rugged and multi-modal, especially when the energy is derived from a neural network (Montúfar et al., 2014). This leads to slow mixing for MCMC, even for gradient-based methods such as Langevin sampling (Roberts & Tweedie, 1996; Welling & Teh, 2011). Second, we must sample on the Lie group and respect manifold constraints, while the energy is defined in data space rather than directly on the group. The first issue motivates the use of diffusion sampling methods, which have shown promise in sampling efficiently from Boltzmann densities. Fig. 2 illustrates the benefits of diffusion. However, as we will see, existing methods do not address the second issue for general Lie groups.

### 4.2. Diffusion sampling with trivialized target score

We introduce Transformation-Inverting Energy Diffusion (TIED), and its associated diffusion sampler on Lie groups, overcoming the aforementioned challenges. To simplify the notation, we fix a datum $\tilde{\mathbf{x}}$ and write the transformation inversion posterior defined in §3.2 as $p(g) \equiv p(g \mid \tilde{\mathbf{x}})$. Consider the energy $E : G \to \mathbb{R}$ associated with the posterior:

$$E(g) \equiv E_{\mathbf{x}}(g^{-1} \cdot \tilde{\mathbf{x}}) - \log |\det J_{g^{-1}}(\tilde{\mathbf{x}})|. \quad (4)$$

To sample from the posterior, we adopt *diffusion sampling*. The idea is to construct a *forward* noising process $p_t(g_t)$ on the group that gradually transforms $p_0 \equiv p$ into a simple noise distribution $p_1$. We then run this process *backwards in time* to turn noise samples into samples from the posterior.

For the forward process, we propose to use a direct Lie-group analogue of the Euclidean variance-exploding SDE in (1).

The idea is to draw infinitesimal perturbations in the Lie algebra $\mathfrak{g}$ and then transport them to the current point on the group via right multiplication (i.e., trivialization):

$$\mathrm{d}g_t = \mathrm{d}(L_{g_t})_e\big[\gamma(t)\,\mathrm{d}\mathbf{w}_t^{\mathfrak{g}}\big], \quad g_0 \sim p, \quad (5)$$

where $\mathrm{d}\mathbf{w}_t^{\mathfrak{g}}$ is Brownian motion in the Lie algebra. Intuitively, we draw a small random step in $\mathfrak{g}$, then use the tangent map $\mathrm{d}(L_g)_e : \mathfrak{g} \to T_g G$ to translate that step from the identity to the current location $g_t$ on the group. This plays the same role as adding Gaussian noise in the Euclidean setting. Visual illustrations can be found in Fig. 2, along with Fig. 6 in §A.

A convenient property of this SDE is that its solution can be written in a simple *multiplicative* form: $g_t = g_0 w_t$, where $w_t \sim k_t$ is a noise random variable on the group that is independent of $g_0$. This mirrors the Euclidean relationship $\mathbf{x}_t = \mathbf{x}_0 + \mathbf{w}_t$ for Gaussian noise $\mathbf{w}_t$, except that on a Lie group we combine the signal and noise via group multiplication rather than vector addition. While $k_t$ does not always admit a closed-form expression due to the group geometry, we can still sample from it efficiently using the exponential map, which is sufficient for our purposes (see §B.9).

For diffusion sampling, we reverse this construction: we start from noise samples $w_1 \sim k_1$ and run the SDE backwards in time to obtain samples from $p_0 \equiv p$. This is valid when the diffusion strengths $\gamma(t)$ are large enough that $k_1$ approximates the marginal $p_1$ of the forward process. We prove that this time-reversed dynamics is again a trivialized diffusion: it evolves on the group via Lie-algebra noise, but now with an additional drift term given by a *trivialized score* (score evaluated in the Lie algebra).

**Proposition 4.1** (Reverse trivialized SDE). *If each $p_t(g_t)$ is smooth and positive with respect to the Haar measure, then the time-reversal of (5) is*

$$\mathrm{d}g_t = \mathrm{d}(L_{g_t})_e \left[ -\gamma(t)^2 \, \nabla_{\mathfrak{g}} \log p_t(g_t) \, \mathrm{d}t + \gamma(t) \, \mathrm{d}\bar{\mathbf{w}}_t^{\mathfrak{g}} \right], \quad (6)$$

*where $\bar{\mathbf{w}}_t^{\mathfrak{g}}$ is Brownian motion in $\mathfrak{g}$ run backwards in time, and $\nabla_{\mathfrak{g}} \log p_t(g_t)$ is the trivialized score, i.e., the score expressed as an element of the Lie algebra.*

A proof is in §B.5, which derives and compares the density evolutions induced by the forward and reverse processes. If we can evaluate or estimate the trivialized score, diffusion sampling becomes straightforward: we discretize the reverse-time SDE and update group elements using the exponential map at each step. We present the resulting sampling scheme in §B.9 and focus next on how to estimate the score.

A natural first idea for estimating the score is to learn it with a neural network via score matching (Huang et al., 2022; De Bortoli et al., 2022; Zhu et al., 2025). However, these methods typically assume access to *clean* samples from $p_0 \equiv p$. In our setting, the posterior $p$ is only available through its associated energy $E \equiv -\log p$. To address this problem, we introduce a new estimator of the trivialized score that uses the energy instead of requiring clean samples.

We build on the work of Akhound-Sadegh et al. (2024); De Bortoli et al. (2024), who use the *target score identity* to express scores of noisy distributions in terms of gradients of a clean energy. We present a new generalization of this identity to Lie groups that is compatible with trivialization:

**Proposition 4.2** (Trivialized target score identity). *For the forward SDE in (5), we have*

$$\nabla_{\mathfrak{g}} \log p_t(g_t) = \int_G \nabla_{\mathfrak{g}} \log p_0(g_0) \, p_{0|t}(g_0 \mid g_t) \mathrm{d}\mu(g_0) \quad (7)$$

*where the argument of $\log p_0$ is interpreted as $g_t b$ for $b \equiv g_t^{-1} g_0$, and $\nabla_{\mathfrak{g}}$ is taken with respect to $g_t$.*

A proof is in §B.6, which writes $p_t$ as a convolution of the clean density $p_0$ and the noise kernel $p_{t|0}$, then takes the trivialized gradient followed by the log-derivative trick. The result shows that the trivialized score at time $t$ can be written as an *average* of the initial score $\nabla_{\mathfrak{g}} \log p_0$ over all possible starting points $g_0$ that could have led to the current state $g_t$, weighted by the conditional density $p_{0|t}$. Since $\nabla_{\mathfrak{g}} \log p_0 = -\nabla_{\mathfrak{g}} E$ and we can access $E$, this gives us a way to express the desired score entirely in terms of clean energy gradients.

A key feature of our result is that, unlike the Lie-group extension of the target score identity in De Bortoli et al. (2024), it applies to *general* Lie groups and does not require a bi-invariant metric. The reason is in the use of trivialization. By always expressing gradients as elements of the Lie algebra, we can meaningfully average energy gradients coming from

different points on the group without explicitly transporting vectors between different tangent spaces. This removes the need for the right-invariance assumption used in De Bortoli et al. (2024, §A.4) to handle such transports. We remark that a Lie group admits a bi-invariant metric iff it is isomorphic to $\mathbb{R}^k \times K$, with $K$ compact (Milnor, 1976, Lem. 7.5). This is a strong constraint. Many practical groups fail this, including the Euclidean group, the general linear group in $d > 1$, the affine group, and the projective group. Many Lie point symmetries of PDEs, relevant to scientific applications, also fall outside. By contrast, our result broadly applies to these practical groups, as we demonstrate in our experiments.

Based on the identity, we now derive a form of the trivialized score that is suitable for Monte Carlo estimation. The idea is to rewrite (7), currently weighted by the conditional density $p_{0|t}(g_0 \mid g_t)$, into an average weighted by the noise density $k_t$ which can be sampled easily. We achieve this by exploiting the multiplicative form $g_t = g_0 w$ for $w \sim k_t$, and performing a corresponding change of variables on the group. We leave the derivation in §B.7, which is based on a nontrivial extension of the Euclidean case in Akhound-Sadegh et al. (2024) to Lie groups.

**Proposition 4.3** (Monte Carlo score estimator). *For the forward SDE in (5), where $p_0$ is a Boltzmann density specified by a smooth energy $E$, we have*

$$\nabla_{\mathfrak{g}} \log p_t(g_t) = \nabla_{\mathfrak{g}} \log \int_G k_t(w) \, h(g_t, w) \, \mathrm{d}\mu(w),$$
$$h(g_t, w) = \exp\!\left(-E(g_t w^{-1}) - \log \lambda(w)\right) \quad (8)$$

*where $\lambda$ accounts for the change of the Haar measure under inversion, $\mathrm{d}\mu(w^{-1}) = \lambda(w)^{-1}\mathrm{d}\mu(w)$.*

A proof is in §B.7, showing that substituting the Boltzmann form $p_0 \propto e^{-E}$ into the convolution form of $p_t$ cancels out the normalizing constant in the score, leaving an expectation over the noise kernel $k_t$. This expression is well-suited for practical estimation. We can efficiently draw samples $w \sim k_t$, and approximate (8) via Monte Carlo. We describe this procedure in detail and prove its consistency in §B.8.

### 4.3. Practical implementation

Alg. 1 presents a practical implementation of energy-based diffusion sampling with TIED. It runs a time-discretized reverse SDE on the Lie group using energy-based MC estimation of the trivialized score. The number of samples $N$ trades off computational cost and the quality of estimation. In our experiments, we find that $N \lesssim 10$ usually yields satisfactory performance and can be parallelized. The implementation relies on a nice property of the score estimator that all its required components, including the volume correction $\lambda$ and the trivialized gradient $\nabla_{\mathfrak{g}}$, can be computed for general Lie groups using standard automatic differentiation tools. We explain the computations in detail in §B.9 and §B.10.

**Algorithm 1** Sampling with TIED

**Input:** Lie group $G$, energy $E$, noise schedule $\gamma$, step size $\Delta t$, Monte Carlo sample size $N$

**Output:** Sample $\hat{g}_0 \sim p \propto e^{-E(\cdot)}$

1 $M \leftarrow 1/\Delta t$          `// number of steps`
2 $\hat{g}_M \sim k_1$ using (24)        `// initialization`
3 **for** $m \leftarrow M$ **to** 1 **do**
4     $t \leftarrow m\Delta t$            `// current time`
       `// trivialized score estimation`
5     **for** $i \leftarrow 1$ **to** $N$ **do**
6        $w^{(i)} \sim k_t$ using (24)      `// MC sample`
7        $l^{(i)} \leftarrow \log \lambda(w^{(i)})$; Prop. B.8    `// correction`
8     **end**
9     $f(\cdot) \leftarrow \log \sum_i e^{-E\left((\cdot) w^{(i)-1}\right) - l^{(i)}}$    `// logsumexp`
10    $\hat{s}_m \leftarrow \nabla_{\mathfrak{g}} f(\hat{g}_m)$; Prop. A.1      `// autograd`
       `// reverse diffusion step`
11    $\bar{z}_m \sim \mathcal{N}(0, \Delta t I)$          `// noise`
12    $\hat{g}_{m-1} \leftarrow \hat{g}_m \exp(\gamma(t)^2 \hat{s}_m \Delta t + \gamma(t)\bar{z}_m)$   `// update`
13 **end**

## 5. Experiments

We evaluate TIED on (i) a synthetic sampling problem on a high-dimensional Lie group; (ii) two image classification problems using a trained convolutional neural network under unknown affine and homography (perspective) transformations; and (iii) two partial differential equations (PDEs) solving problems using trained neural operators under unknown Lie point symmetry transformations. Efficiency, cost-accuracy tradeoff, and ablation analysis can be found in §C.

### 5.1. Synthetic sampling task on $\mathrm{SO}(10)$

The first setting we consider is a synthetic energy-based sampling problem on a high-dimensional Lie group. We adopt the setup of Kong & Tao (2024) that considers the special orthogonal group $\mathrm{SO}(10)$ represented as a set of $10 \times 10$ matrices $\mathbf{X}$ with determinant 1. This group is chosen since its large intrinsic dimension $\dim \mathrm{SO}(10) = 45$ offers a challenge for sampling methods (in contrast to e.g. $\dim \mathrm{SO}(3) = 3$). The energy we consider is $E : \mathbf{X} \mapsto -10\mathbf{X}_{1,1}^2$ where $\mathbf{X}_{1,1}$ is the value of the top-left matrix element. This energy induces a multimodal density $p \propto e^{-E(\cdot)}$ on the group, thereby offering a testbed aligned with our motivations in §4.1.

We compare TIED against the trivialized kinetic Langevin sampler proposed in Kong & Tao (2024), which only uses the energy gradient $\nabla_{\mathfrak{g}} E$. We run the Langevin sampler for 100,000 steps to ensure convergence. For TIED, we run 100 steps of diffusion, with sample size 100 for MC estimation of the score $\nabla_{\mathfrak{g}} \log p_t$. We provide the results in Fig. 3, focusing on the top-left matrix element $\mathbf{X}_{1,1}$ for visualization. The results show that TIED samples from the correct multimodal

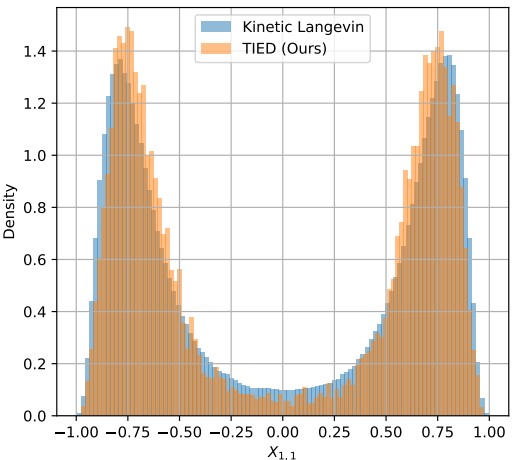

*(a)* Distributions of sampled $\mathbf{X}_{1,1}$.

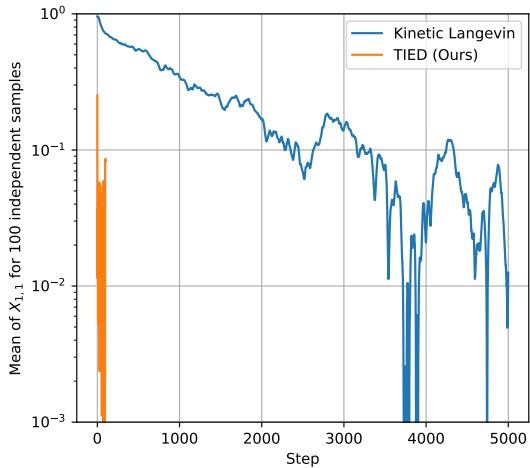

*(b)* Mean of $\mathbf{X}_{1,1}$ over timesteps (lower the better).

*Figure 3.* Sampling on $\mathrm{SO}(10)$ under energy $E : \mathbf{X} \mapsto -10\mathbf{X}_{1,1}^2$ using a kinetic Langevin sampler (Kong & Tao, 2024) and TIED (Ours). The distribution of $\mathbf{X}_{1,1}$ induced by the energy has two symmetric modes around zero, and thus the mean of $\mathbf{X}_{1,1}$ approaches zero as the sampling converges.

distribution, and does so in far fewer sampling steps, thanks to the diffusion formulation.

### 5.2. Affine/homography invariant image classification

Next, we demonstrate TIED on image classification problems using a trained convolutional neural network under unknown affine and homography (perspective) transformations, closely following the experimental setup of Shumaylov et al. (2024). As our pretrained network $f_\theta$, we use a ResNet18 (He et al., 2016) trained to classify $40 \times 40$ padded MNIST images. To test its robustness and generalization, we consider two challenging transformation groups. The first is the group of

*Table 1.* MNIST classification test accuracy and FID over five seeds.

| dataset | MNIST | | | |
|---|---|---|---|---|
| test transformations | none | | | |
| | Acc. (%) | | FID | |
| ResNet18 | 99.35 | | - | |
| test transformations | $\text{Aff}(2, \mathbb{R})$ | | $\text{PGL}(3, \mathbb{R})$ | |
| | Acc. (%) | FID | Acc. (%) | FID |
| affConv / homConv * | 95.08 | - | 95.71 | - |
| ResNet18 | 55.48 | - | 87.95 | - |
| Energy: VAE evidence lower bound (+ adv. reg.) | | | | |
| ResNet18 + ITS | 45.97±0.39 | 10.89 | n/a | n/a |
| ResNet18 + FoCal | 86.38±0.00 | 3.39 | 89.69±0.00 | 2.20 |
| ResNet18 + Langevin | 74.72±0.29 | 7.15 | 93.75±0.10 | 0.77 |
| ResNet18 + LieLAC | 94.38±0.17 | 0.91 | 97.39±0.08 | 0.58 |
| ResNet18 + TIED (Ours) | **96.95±0.12** | **0.71** | **97.43±0.06** | **0.57** |
| Energy: Classifier logit confidence | | | | |
| ResNet18 + ITS | 67.18±0.14 | 9.90 | n/a | n/a |
| ResNet18 + FoCal | 66.97±0.00 | 10.92 | 86.45±0.00 | 3.89 |
| ResNet18 + Langevin | 53.98±0.11 | 15.29 | 88.25±2.48 | 2.48 |
| ResNet18 + LieLAC | 73.79±0.08 | 7.03 | 85.93±0.17 | 2.66 |
| ResNet18 + TIED (Ours) | **82.64±0.11** | **5.05** | **89.84±0.17** | **1.83** |

* from the original paper's table



| Distorted | Inverted | Distorted | Inverted |
|---|---|---|---|
| Class 7 | Class 2 | Class 2 | Class 3 |
| Class 1 | Class 3 | Class 7 | Class 4 |
| Class 2 | Class 5 | Class 1 | Class 7 |
| Class 1 | Class 7 | Class 7 | Class 8 |

*(a)* $\text{Aff}(2, \mathbb{R})$ *(b)* $\text{PGL}(3, \mathbb{R})$

*Figure 4.* For each transformation group, we show cases of misclassification by ResNet18 $f_\theta$. From left: $f_\theta$ prediction, $f_\theta$ prediction under test-time invariance via TIED (Ours) using classifier energy.

affine transformations $\text{Aff}(2, \mathbb{R})$, and the second is the group of homography (perspective) transformations, isomorphic to the projective general linear group $\text{PGL}(3, \mathbb{R})$. Despite their widespread use in computer vision, these groups have high mathematical complexities: both are noncompact, non-Abelian, and lack a bi-invariant metric. We construct the respective test sets as 10,000-sized random subsets of affNIST and homNIST test images from MacDonald et al. (2022). In addition to classification accuracy of $f_\theta$, we measure the Fréchet inception distance (FID) (Heusel et al., 2017; Fatir, 2018) between the inverse-transformed test images and the training images of $f_\theta$ as a supplementary metric.

The baselines include general optimization and sampling methods on Lie groups, as well as specialized methods for inverting transformations for robust neural perception. For the former, we test trivialized kinetic Langevin (Kong & Tao,

2024) and LieLAC (Shumaylov et al., 2024), which respectively perform trivialized gradient-based sampling and optimization (Lezcano Casado, 2019). For the latter, we test ITS (Schmidt & Stober, 2024), which performs an iterative search specifically designed for affine transforms, and FoCal (Singhal et al., 2025), which performs Bayesian optimization on transformation spaces (we use the Lie algebra).

Notably, all the baselines use an energy function (sometimes referred to as a data prior) to identify an inverse transformation, making it natural to compare them under the same choices of energy. For image classification, we experiment with two choices of energy, broadly representative of probabilistic and predictive ones. For the probabilistic energy, we use the ELBO of a VAE trained on clean MNIST images augmented with adversarial regularization, adopted from Shumaylov et al. (2024). For the predictive energy, we use a confidence-based energy measured by $-\texttt{logsumexp}$ of output logits (Grathwohl et al., 2020), similarly to Schmidt & Stober (2024); Singhal et al. (2025). For this, we use the same ResNet18 $f_\theta$ for measuring the classification performance.

The results are in Tab. 1. While trained ResNet18 achieves high accuracy on clean images, affine or homography transformations substantially degrade it. Kinetic Langevin and LieLAC are based on energy gradients. Both restore the accuracy reasonably well, but relatively underperform on affine using classifier energy. This is possibly due to the large magnitudes of transformations (as evidenced by the ResNet18 accuracies) and anomalies in the classifier's energy landscape affecting the gradients. ITS and FoCal, which are iterative search methods, attain decent results on affine with classifier energy. We conjecture that their nonlocal iterative exploration helps handle the anomalies in the energy landscape. Finally, our method, TIED, combines precise local optimization via gradients with exploration by diffusion at high noise levels. As a result, it outperforms all baselines in all settings, significantly improving the accuracy under affine transforms and classifier energy (55.48% → 82.64%). We emphasize that, with confidence energy, the classifier is essentially self-correcting its predictions (Fig. 4). With a proper choice of the energy, TIED also outperforms specialized equivariant networks affConv and homConv (MacDonald et al., 2022).

### 5.3. Point symmetry equivariant PDE solving

We test TIED on a challenging problem of solving partial differential equations (PDEs) with a trained neural operator under unknown Lie point symmetry transformations. Here, the model $f_\theta$ performs function-valued regression. The input is a function evaluation $\mathbf{u}_0$ at a set of space and time points $(\mathbf{x}_0, t_0)$, which specifies an initial condition. The task is to evaluate the function at another set of points $(\mathbf{x}_f, t_f)$, after $\mathbf{u}_0(\mathbf{x}_0, t_0)$ has evolved under the PDE of interest. More details can be found in Li et al. (2020); Lu et al. (2021).

*Table 2.* PDE solving relative test L2 error.

| PDE | 1D Heat Eq. | 1D Heat Eq. + Data Aug. | 1D Burgers Eq. |
|---|---|---|---|
| Test transformations | none | none | none |
| DeepONet | 0.011 | 0.031 | 0.017 |
| Test transformations | $SL(2, \mathbb{R}) \ltimes H(1, \mathbb{R})$ | $SL(2, \mathbb{R}) \ltimes H(1, \mathbb{R})$ | $SL(2, \mathbb{R}) \ltimes (\mathbb{R}^2, +)$ |
| DeepONet | 0.690 | 0.081 | 0.867 |
| Energy: Distance to training domain | | | |
| DeepONet + FoCal | $1.948 \pm 0.608$ | $1.000 \pm 0.355$ | $0.202 \pm 0.017$ |
| DeepONet + Kinetic Langevin | $0.565 \pm 0.046$ | $0.247 \pm 0.059$ | $0.739 \pm 0.019$ |
| DeepONet + LieLAC | $0.066 \pm 0.019$ | $0.078 \pm 0.017$ | $0.187 \pm 0.004$ |
| DeepONet + TIED (Ours) | $\mathbf{0.043 \pm 0.002}$ | $\mathbf{0.052 \pm 0.001}$ | $\mathbf{0.178 \pm 0.013}$ |

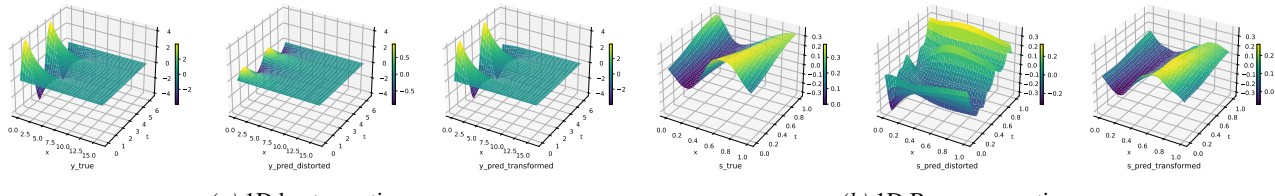

*(a)* 1D heat equation.      *(b)* 1D Burgers equation.

*Figure 5.* For each PDE, we show an out-of-domain case for DeepONet $f_\theta$. From left: true solution, $f_\theta$ prediction, $f_\theta$ prediction under test-time equivariance via TIED (Ours). **Zoom in** for a better view.

Many PDEs possess a symmetry $G$ that transforms a solution $\mathbf{u}(\mathbf{x}, t)$, to create other valid solutions (Olver, 1993; Brandstetter et al., 2022; Akhound-Sadegh et al., 2023). Since a neural operator $f_\theta$ is trained for a specific range of initial conditions $\mathbf{u}_0(\mathbf{x}_0, t_0)$ and prediction points $(\mathbf{x}_f, t_f)$, one expects it to fail outside the training domain. Akin to our problem setup, this can be overcome by finding a symmetry transformation $g$ which acts on the input as $g^{-1} \cdot (\mathbf{u}_0(\mathbf{x}_0, t_0), \mathbf{x}_f, t_f)$ to push it into the training domain of $f_\theta$. Then, $f_\theta$ can make out-of-domain predictions as $g \cdot f_\theta(g^{-1} \cdot (\mathbf{u}_0(\mathbf{x}_0, t_0), \mathbf{x}_f, t_f))$. A challenge is the complexity of Lie point symmetries: in many cases, they are noncompact and non-Abelian. This makes TIED an appealing candidate as it handles these cases.

We closely follow the setup of Shumaylov et al. (2024), using DeepONet neural operators (Lu et al., 2021) as the pretrained $f_\theta$. To test the robustness and generalization, we use two PDEs with challenging symmetry groups. The first is the 1D heat equation $u_t - \nu u_{xx} = 0$ with the symmetry group $SL(2, \mathbb{R}) \ltimes H(1, \mathbb{R})$, the semidirect product of special linear group and rank-one polarized Heisenberg group. The second is the 1D Burgers' equation $u_t + u u_x - \nu u_x x = 0$ with the symmetry group $SL(2, \mathbb{R}) \ltimes (\mathbb{R}^2, +)$. The test sets are constructed by applying these transformations to send $\mathbf{u}_0$ out of the training domain of $f_\theta$. For the choice of energy, we follow Shumaylov et al. (2024) and use a distance measure between the input $(\mathbf{u}_0(\mathbf{x}_0, t_0), \mathbf{x}_f, t_f)$ and the training domain of $f_\theta$.

The results are in Tab. 2 and Fig. 5. While all DeepONets excel in their training domain, their performance degrades when presented with out-of-domain transformations, with data augmentation reducing the degradation but not perfectly. While FoCal performs poorly on heat equations, it has a strong performance on Burgers equation. We conjecture that this is partially due to the Euclidean component $(\mathbb{R}^2, +)$ in the semidirect product, which may have created a more amenable environment for Bayesian optimization. Although both are based on gradients, LieLAC outperforms kinetic Langevin. This could be due to the Brownian motion term slowing down convergence. Our method TIED can leverage smoothed energy landscape at high noise levels effectively to speed up convergence, and achieves the best performance in all cases.

## 6. Conclusion

In this work, we studied the problem of inverting unknown data transformations from general Lie groups in a setting where we have access to a *data-space prior* specified by an energy function. Our method TIED transforms a test-time input so that it becomes well aligned with the model's training distribution and thereby achieves strong performance. This setting is increasingly relevant in the era of large foundation models, where robustness to out-of-distribution transformations is a practical concern. TIED operates by reversing a diffusion process over transformations in the Lie algebra. It applies to a large class of groups and does not require any training or finetuning. Avenues for future work include improving the efficiency of the sampler. We could consider, e.g., consistency distillation methods (Song et al., 2023). Applications to other inverse problems, such as image registration, would also prove interesting.

## Acknowledgment

This work was supported in part by the National Research Foundation of Korea (RS-2024-00351212, RS-2024-00436165), the Institute of Information & Communications Technology Planning & Evaluation (IITP) (RS-2024-00509279, RS-2022-II220926, RS-2022-II220959, RS-2019-II190075), the InnoCORE program of the Ministry of Science and ICT (AI Meta-Scientist, N10260110), and the "HPC support" project, funded by the Korean government (MSIT). SR and S-OK would like to acknowledge support by CIFAR, NSERC Discovery and Samsung AI Labs. Computational resources were provided in part by Mila and Compute Canada.

## Impact Statement

This work introduces a probabilistic framework and algorithm for inverting unknown data transformations from general Lie groups, aiming to improve the generalization of pretrained deep neural networks. Our contributions are theoretical and methodological, and we performed evaluations on publicly available benchmarks either in terms of artifacts or details of simulation. We do not anticipate direct ethical risks associated with this approach.

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

# A. Background

We provide an overview of the mathematical background, and refer the readers to Lee (2012); Tu (2010) for more details.

**Lie group**  A Lie group $G$ is a group that is also a smooth manifold, such that multiplications of elements and taking inverses are smooth. In deep learning, it is useful as an abstraction of a class of continuous data transformations such as perspective transformations of an image. For any $g \in G$, we denote by $L_g : G \to G$ the left multiplication map $h \mapsto gh$ and by $R_g : G \to G$ the right multiplication map $h \mapsto hg$. We denote by $\mu$ the left-invariant (unnormalized) Haar measure on $G$.

**Tangent spaces**  As a manifold, a Lie group $G$ has a tangent space $T_g G$ at each point $g$, intuitively containing all possible directions of infinitesimal updates at $g$. We can map between tangent spaces using the notion of differentials. For any smooth function $f : G \to G$ and any $g \in G$, we denote by $\mathrm{d}f_g : T_g G \to T_{f(g)} G$ the differential of $f$ evaluated at $g$. For left multiplications $L_g$, the differentials $\mathrm{d}(L_g)_h : T_h G \to T_{gh} G$ are bijective linear maps between tangent spaces called the tangent maps. The tangent maps satisfy $\mathrm{d}(L_g)_{hk} \circ \mathrm{d}(L_h)_k = \mathrm{d}(L_{gh})_k$ and $\mathrm{d}(L_e)_g = \mathrm{id}_{T_g G}$ for all $g, h, k \in G$.

**Lie algebra**  While a Lie group is a curved manifold, a lot of its properties derive from its tangent space at the identity, the Lie algebra, defined by $\mathfrak{g} \equiv T_e G$. For an $n$-dimensional Lie group, the Lie algebra $\mathfrak{g}$ is an $n$-dimensional vector space equipped with a binary operation $[\cdot, \cdot] : \mathfrak{g} \times \mathfrak{g} \to \mathfrak{g}$ called the Lie bracket. We can always construct an inner product $\langle \cdot, \cdot \rangle_{\mathfrak{g}}$ on $\mathfrak{g}$ by choosing any basis $\{\mathbf{e}_1, ..., \mathbf{e}_n\}$ and defining $\langle \sum_i u_i \mathbf{e}_i, \sum_i v_i \mathbf{e}_i \rangle_{\mathfrak{g}} \equiv \sum_i u_i v_i$. This is the unique inner product on $\mathfrak{g}$ for which the chosen basis is orthonormal. The exponential map $\exp : \mathfrak{g} \to G$ is a smooth map that specifies group structure from the Lie algebra. In general, this specification is local around the identity $e = \exp(\mathbf{0})$ as $\exp$ can be non-surjective for non-compact $G$. Yet, for connected Lie groups, any element can be reached as a finite product of these local elements (Olver, 1993, Prop. 1.24).

**Gradients**  We can define the notion of gradients on a Lie group by equipping it with a choice of metric. For a finite-dimensional Lie group, one possible choice is a metric $\langle \cdot, \cdot \rangle$ that is left-invariant:

$$\langle \mathrm{d}(L_h)_g \mathbf{u}, \mathrm{d}(L_h)_g \mathbf{v} \rangle_{hg} = \langle \mathbf{u}, \mathbf{v} \rangle_g.$$

Such a metric can always be constructed by extending any inner product $\langle \cdot, \cdot \rangle_{\mathfrak{g}}$ on the Lie algebra:

$$\langle \mathbf{u}, \mathbf{v} \rangle_g \equiv \langle \mathrm{d}(L_{g^{-1}})_g \mathbf{u}, \mathrm{d}(L_{g^{-1}})_g \mathbf{v} \rangle_{\mathfrak{g}}.$$

Then, for any smooth function $f : G \to \mathbb{R}$, the gradient $\nabla f(g)$ at each point $g \in G$ is given as the unique element of the tangent space $T_g G$ that satisfies the following for all $\mathbf{v} \in T_g G$:

$$\langle \nabla f(g), \mathbf{v} \rangle_g = \mathrm{d}f_g(\mathbf{v}), \tag{9}$$

where $\mathrm{d}f_g : T_g G \to \mathbb{R}$ is the differential of $f$ evaluated at $g$ (Lee, 2012, §13).

**Trivialization**  In a deep learning context, the curved geometry of Lie groups can be challenging to deal with. We can narrow this down to two specific difficulties in our problem setup.

1. In energy-based diffusion, one usually employs an expectation form of the score function that averages energy gradients evaluated across the state space (De Bortoli et al., 2024; Akhound-Sadegh et al., 2024). However, on a Lie group, gradients of a function at different points live in different tangent spaces and hence are not directly compatible, e.g., cannot be added or averaged.

2. In general, during sampling on a Lie group, the update directions live in tangent spaces that change over time simultaneously as a sample is updated, requiring a careful handling.

We can sidestep both difficulties by always working in the Lie algebra instead of arbitrary tangent spaces. This technique is called (left-)trivialization (Lezcano Casado, 2019; Tao & Ohsawa, 2020; Kong & Tao, 2024; Zhu et al., 2025). We use two related tools that together address the difficulties.

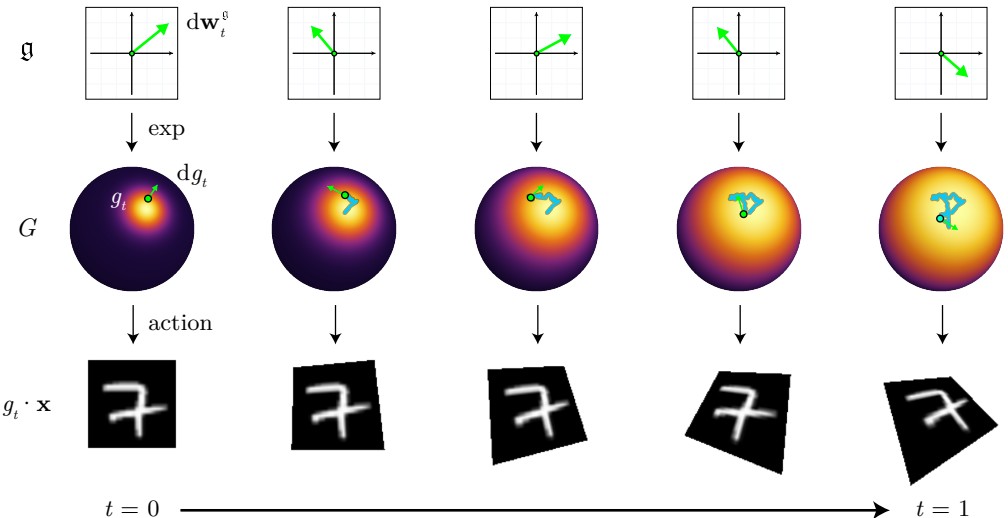

*Figure 6.* An illustration of trivialized forward diffusion on the projective linear group acting on an image.

Our first tool is the **trivialized gradient** that can be understood as gradient evaluated in the Lie algebra. For any smooth function $f : G \to \mathbb{R}$, we define the trivialized gradient $\nabla_{\mathfrak{g}} f(g) \in \mathfrak{g}$ at each point $g \in G$ by mapping the standard gradient $\nabla f(g) \in T_g G$ through the tangent map $\mathrm{d}(L_{g^{-1}})_g : T_g G \to \mathfrak{g}$:

$$\nabla_{\mathfrak{g}} f(g) \equiv \mathrm{d}(L_{g^{-1}})_g \nabla f(g).$$

Since trivialized gradients always live in the same vector space $\mathfrak{g}$, we can add or average them without restrictions. This underlies our novel score identity on general Lie groups.

Another nice property of trivialized gradients is that, despite their definition involving a tangent map, they can be computed directly using automatic differentiation without any explicit handling of tangent maps. This knowledge is not new (Kobilarov, 2014a;b), but rarely used in deep learning contexts.

**Proposition A.1** (Trivialized gradient). *Let $G$ be a finite-dimensional Lie group. Under any choice of a left-invariant metric, for any smooth function $f : G \to \mathbb{R}$, the following holds for all $g \in G$:*

$$\nabla_{\mathfrak{g}} f(g) = \nabla_{\mathbf{v}} f(g \exp(\mathbf{v}))|_{\mathbf{v}=\mathbf{0}} .$$

*Proof.* The proof is given in §B.3. □

Also, as a useful property, we can use the standard log-derivative trick on trivialized gradients:

**Proposition A.2** (Trivialized score). *Let $G$ be a finite-dimensional Lie group. Under any choice of a left-invariant metric, for any positive and smooth function $p : G \to \mathbb{R}$, it holds that $\nabla_{\mathfrak{g}} p = p \nabla_{\mathfrak{g}} \log p$.*

*Proof.* The proof is given in §B.4. □

Our second tool is a **trivialized SDE** that specifies diffusion on Lie groups using components in the Lie algebra. Specifically, for an $n$-dimensional Lie group $G$ and any choice of an orthonormal basis of the Lie algebra $\mathfrak{g}$, we consider Itô SDEs on the group having the following form:

$$\mathrm{d}g_t = \mathrm{d}(L_{g_t})_e \left[ \phi(g_t, t)\, \mathrm{d}t + \gamma(t)\, \mathrm{d}\mathbf{w}_t^{\mathfrak{g}} \right], \tag{10}$$

with $\mathrm{d}\mathbf{w}_t^{\mathfrak{g}}$ the Brownian motion on Lie algebra given as $\mathrm{d}\mathbf{w}_t^{\mathfrak{g}} = \sum_i \mathbf{e}_i \mathrm{d}w_t^i$ where $\mathrm{d}w_t^1, ..., \mathrm{d}w_t^n$ are independent standard Brownian motions on $\mathbb{R}$ and $\{\mathbf{e}_1, ..., \mathbf{e}_n\}$ is the chosen orthonormal basis of $\mathfrak{g}$. In addition, $\phi(\cdot, t) : G \to \mathfrak{g}$ is the drift coefficient, and $\gamma(t) \in \mathbb{R}$ is the diffusion coefficient.

The trivialized nature of the SDE motivates a natural Euler time-discretization scheme for sampling that eliminates the need to handle moving tangent spaces. This is based on the fact that, for any fixed $\mathbf{v} \in \mathfrak{g}$, the trivialized dynamics $\mathrm{d}g = \mathrm{d}(L_g)_e \mathbf{v} \, \mathrm{d}t$ has a closed-form solution $g(t) = g(0) \exp(t\mathbf{v})$ that can be computed using the exponential map (Lee, 2012, (8.13) and Prop. 20.8). Specifically, for step size $\Delta t > 0$, an Euler time-discretization can be written as follows, where $m = 0, ..., 1/\Delta t$:

$$\hat{g}_{m+1} = \hat{g}_m \, \exp\left(\phi(\hat{g}_m, m\Delta t)\,\Delta t + \gamma(m\Delta t)\,\mathbf{z}_m\right), \quad \mathbf{z}_m \sim \mathcal{N}(\mathbf{0}, \Delta t\mathbf{I}), \quad \hat{g}_0 \overset{d}{=} g_0. \tag{11}$$

In Fig. 6, we show an illustration of trivialized diffusion on the projective linear group acting on an image, with zero drift and diffusion coefficient one.

## B. Proofs

### B.1. Proof of Proposition 3.1

**Proposition 3.1** (Transformation inversion posterior). *Let $G$ act freely on $\mathcal{X}$. Then,*

1. *The posterior distribution of g has density $p\left(g \mid \tilde{\mathbf{x}}\right) \propto p_\mathbf{x}\left(g^{-1} \cdot \tilde{\mathbf{x}}\right)|\det J_{g^{-1}}\left(\tilde{\mathbf{x}}\right)|.$*

2. *The random variable $\mathbf{x}' = h^{-1} \cdot \tilde{\mathbf{x}}$, with $h \sim p\left(g \mid \tilde{\mathbf{x}}\right)$ has density $p\left(\mathbf{x}' \mid \tilde{\mathbf{x}}\right) \propto p_\mathbf{x}\left(\mathbf{x}'\right) \mathbb{1}_{G \cdot \tilde{\mathbf{x}}}\left(\mathbf{x}'\right)$ w.r.t. $\mu_{G \cdot \tilde{\mathbf{x}}}.$*

*Proof.* For the first statement, using independence of $\mathbf{x}$ and $g$ and the relation $\tilde{\mathbf{x}} = g \cdot \mathbf{x}$, we have

$$p\left(\mathbf{x}, \tilde{\mathbf{x}} \mid g\right) = p_\mathbf{x}\left(\mathbf{x}\right) p\left(\tilde{\mathbf{x}} \mid \mathbf{x}, g\right)$$
$$p\left(\mathbf{x}, \tilde{\mathbf{x}} \mid g\right) = p_\mathbf{x}\left(\mathbf{x}\right) \delta\left(\tilde{\mathbf{x}} - g \cdot \mathbf{x}\right)$$

Marginalizing with respect to $\mathbf{x}$ yields

$$p\left(\tilde{\mathbf{x}} \mid g\right) = \int p_\mathbf{x}\left(\mathbf{x}\right) \delta\left(\tilde{\mathbf{x}} - g \cdot \mathbf{x}\right) \mathrm{d}\mathbf{x}$$

Since the group action is diffeomorphic, we have by the change-of-variables formula

$$p\left(\tilde{\mathbf{x}} \mid g\right) = \int p_\mathbf{x}\left(\mathbf{x}\right) \delta\left(g^{-1} \cdot \tilde{\mathbf{x}} - \mathbf{x}\right) |\det J_{g^{-1}}\left(\tilde{\mathbf{x}}\right)| \mathrm{d}\mathbf{x}$$
$$p\left(\tilde{\mathbf{x}} \mid g\right) = p_\mathbf{x}\left(g^{-1} \cdot \tilde{\mathbf{x}}\right) |\det J_{g^{-1}}\left(\tilde{\mathbf{x}}\right)|$$

With a uniform prior on $g$, we obtain via Bayes' rule

$$p\left(g \mid \tilde{\mathbf{x}}\right) \propto p_\mathbf{x}\left(g^{-1} \cdot \tilde{\mathbf{x}}\right) |\det J_{g^{-1}}\left(\tilde{\mathbf{x}}\right)|$$

For the second statement, for $\mathbf{x}' = g^{-1} \cdot \tilde{\mathbf{x}}$ and measurable $B \subseteq G \cdot \tilde{\mathbf{x}}$, we have

$$\mathbb{P}\left(\mathbf{x}' \in B \mid \tilde{\mathbf{x}}\right) = \int_{g \in G} p\left(g \mid \tilde{\mathbf{x}}\right) \mathbb{1}_B\left(g^{-1} \cdot \tilde{\mathbf{x}}\right) \mathrm{d}\mu(g)$$
$$\mathbb{P}\left(\mathbf{x}' \in B \mid \tilde{\mathbf{x}}\right) \propto \int_{g \in G} p_\mathbf{x}\left(g^{-1} \cdot \tilde{\mathbf{x}}\right) |\det J_{g^{-1}}\left(\tilde{\mathbf{x}}\right)| \mathbb{1}_B\left(g^{-1} \cdot \tilde{\mathbf{x}}\right) \mathrm{d}\mu(g)$$

Making the change of variable $\mathbf{x}' = g^{-1} \cdot \tilde{\mathbf{x}}$ and using the fact that the group action is diffeomorphic, we have by the change of variable formula

$$\mathbb{P}\left(\mathbf{x}' \in B \mid \tilde{\mathbf{x}}\right) \propto \int_{\mathbf{x}' \in G \cdot \tilde{\mathbf{x}}} p_\mathbf{x}\left(\mathbf{x}'\right) \mathbb{1}_B\left(\mathbf{x}'\right) \mathrm{d}\mu_{G \cdot \tilde{\mathbf{x}}}(\mathbf{x}')$$

Therefore we have the following density

$$p\left(\mathbf{x}' \mid \tilde{\mathbf{x}}\right) \propto p_\mathbf{x}\left(\mathbf{x}'\right) \mathbb{1}_{G \cdot \tilde{\mathbf{x}}}\left(\mathbf{x}'\right)$$

with respect to $\mu_{G \cdot \tilde{\mathbf{x}}}.$ $\qquad\square$

**On the free action assumption** The assumption that $G$ acts freely is mainly a technical condition used for the proof. In practice, TIED does not require freeness, as it only relies on defining and sampling an energy over $G$. When the action is not free, the main effect is redundancy: multiple group elements correspond to the same transformed sample. This does not affect recovery of $g^{-1} \cdot \tilde{\mathbf{x}}$, which is invariant along stabilizer directions.

While noncompact stabilizers can lead to a non-normalizable posterior over $G$, where the density is not well-defined mathematically, even in this situation, the sampling scheme is still well-defined, and the induced sample $g^{-1} \cdot \tilde{\mathbf{x}}$ remains unchanged along those directions, so this does not impact recovery in practice. Although the sampler may not visit all modes, it is not a problem in practice because the different modes are associated with identical copies of the data.

Finally, in the applications we consider, the freeness assumption holds. It is violated when the input data has perfect symmetry, which would be artificial. Overall, the freeness assumption simplifies the theory, but is not a limitation in practice.

### B.2. Proof of Proposition 3.2

**Proposition 3.2** (Equivariance of the posterior). *For any $E_{\mathbf{x}} : \mathcal{X} \to \mathbb{R}$, the posterior density $p\left(g \mid \tilde{\mathbf{x}}\right)$ is $G$-equivariant.*

*Proof.* By left-invariance of the Haar measure, the density $p\left(g \mid \tilde{\mathbf{x}}\right)$ is equivariant if

$$p\left(h \cdot g \mid h \cdot \tilde{\mathbf{x}}\right) = p\left(g \mid \tilde{\mathbf{x}}\right), \quad \forall g, h \in G, \tilde{\mathbf{x}} \in \mathcal{X}$$

We can verify this explicitly for the posterior given by

$$p\left(g \mid \tilde{\mathbf{x}}\right) = \frac{e^{-E_{\mathbf{x}}\left(g^{-1} \cdot \tilde{\mathbf{x}}\right)} |\det J_{g^{-1}}\left(\tilde{\mathbf{x}}\right)|}{Z\left(\tilde{\mathbf{x}}\right)}$$

We have

$$p\left(h \cdot g \mid h \cdot \tilde{\mathbf{x}}\right) = \frac{e^{-E_{\mathbf{x}}\left(g^{-1} \cdot h^{-1} \cdot h \cdot \tilde{\mathbf{x}}\right)} |\det J_{(h \cdot g)^{-1}}\left(h \cdot \tilde{\mathbf{x}}\right)|}{Z\left(h \cdot \tilde{\mathbf{x}}\right)}$$

$$p\left(h \cdot g \mid h \cdot \tilde{\mathbf{x}}\right) = \frac{e^{-E_{\mathbf{x}}\left(g^{-1} \cdot \tilde{\mathbf{x}}\right)} |\det J_{(h \cdot g)^{-1}}\left(h \cdot \tilde{\mathbf{x}}\right)|}{Z\left(h \cdot \tilde{\mathbf{x}}\right)}$$

Using the chain rule for the Jacobian, we obtain

$$p\left(h \cdot g \mid h \cdot \tilde{\mathbf{x}}\right) = \frac{e^{-E_{\mathbf{x}}\left(g^{-1} \cdot \tilde{\mathbf{x}}\right)} |\det J_{g^{-1}}\left(\tilde{\mathbf{x}}\right)| |\det J_{h^{-1}}\left(h \cdot \tilde{\mathbf{x}}\right)|}{Z\left(h \cdot \tilde{\mathbf{x}}\right)}$$

We can absorb the factor $|\det J_{h^{-1}}\left(h \cdot \tilde{\mathbf{x}}\right)|$ into the normalization constant by using the chain rule for Jacobians, then a change of variable $g' = h^{-1}g$ followed by left-invariance of the Haar measure

$$\begin{aligned}
\frac{|\det J_{h^{-1}}(h \cdot \tilde{\mathbf{x}})|}{Z(h \cdot \tilde{\mathbf{x}})} &= \frac{|\det J_{h^{-1}}(h \cdot \tilde{\mathbf{x}})|}{\int_G e^{-E_{\mathbf{x}}(g^{-1}h \cdot \tilde{\mathbf{x}})} |\det J_{g^{-1}}(h \cdot \tilde{\mathbf{x}})|\, d\mu(g)} \\
&= \frac{|\det J_{h^{-1}}(h \cdot \tilde{\mathbf{x}})|}{\int_G e^{-E_{\mathbf{x}}(g^{-1}h \cdot \tilde{\mathbf{x}})} |\det J_{g^{-1}h}(\tilde{\mathbf{x}})| |\det J_{h^{-1}}(h \cdot \tilde{\mathbf{x}})|\, d\mu(g)} \\
&= \frac{1}{\int_G e^{-E_{\mathbf{x}}(g'^{-1} \cdot \tilde{\mathbf{x}})} |\det J_{g'^{-1}}(\tilde{\mathbf{x}})|\, d\mu(hg')} \\
&= \frac{1}{Z(\tilde{\mathbf{x}})}
\end{aligned}$$

This gives us $p\left(h \cdot g \mid h \cdot \tilde{\mathbf{x}}\right) = p\left(g \mid \tilde{\mathbf{x}}\right)$. $\qquad\square$

### B.3. Proof of Proposition A.1

**Proposition B.1** (Trivialized gradient). *Let $G$ be a finite-dimensional Lie group. Under any choice of a left-invariant metric, for any smooth function $f : G \to \mathbb{R}$, the following holds for all $g \in G$:*

$$\nabla_{\mathfrak{g}} f(g) = \nabla_{\mathbf{v}} f(g \exp(\mathbf{v}))|_{\mathbf{v}=\mathbf{0}}. \tag{12}$$

*Proof.* Let $\mathbf{u} \in \mathfrak{g}$ be an arbitrary unit vector and let $\gamma(t) \equiv g \exp(t\mathbf{u})$ be an associated smooth curve on the group parameterized by $t \in \mathbb{R}$. Then from Lee (2012, Corollary 3.25) the following holds

$$\mathrm{d}f_g(\dot{\gamma}(0)) = \left. \frac{\mathrm{d}}{\mathrm{d}t} f(\gamma(t)) \right|_{t=0}$$

Starting from the left-hand side, we get

$$\begin{aligned}
\mathrm{d}f_g(\dot{\gamma}(0)) &= \langle \nabla f(g), \dot{\gamma}(0) \rangle_g \\
&= \langle \nabla f(g), \mathrm{d}(L_g)_e \mathbf{u} \rangle_g \\
&= \langle \mathrm{d}(L_{g^{-1}})_g \nabla f(g), \mathrm{d}(L_{g^{-1}})_g \circ \mathrm{d}(L_g)_e \mathbf{u} \rangle_e \\
&= \langle \nabla_{\mathfrak{g}} f(g), \mathbf{u} \rangle_e
\end{aligned}$$

where we use $\dot{\gamma}(0) = \mathrm{d}(L_{\gamma(0)})_e \mathbf{u} = \mathrm{d}(L_g)_e \mathbf{u}$ (Lee, 2012, §8, §9, §20) for the second equality.

Starting from the right-hand side, we get

$$\begin{aligned}
\left. \frac{\mathrm{d}}{\mathrm{d}t} f(\gamma(t)) \right|_{t=0} &= \left. \frac{\mathrm{d}}{\mathrm{d}t} f(g \exp(\mathbf{0} + t\mathbf{u})) \right|_{t=0} \\
&= \nabla_{\mathbf{u}} (f \circ L_g \circ \exp)(\mathbf{0}) \\
&= \langle \nabla_{\mathbf{v}} f(g \exp(\mathbf{v}))|_{\mathbf{v}=\mathbf{0}}, \mathbf{u} \rangle_e
\end{aligned}$$

where, for the second equality, we use the definition of directional derivative of the composite function $f \circ L_g \circ \exp : \mathfrak{g} \to \mathbb{R}$ along the direction $\mathbf{u}$ evaluated at $\mathbf{0}$.

Therefore, for arbitrary unit vector $\mathbf{u} \in \mathfrak{g}$ it holds that

$$\langle \nabla_{\mathfrak{g}} f(g), \mathbf{u} \rangle_e = \langle \nabla_{\mathbf{v}} f(g \exp(\mathbf{v}))|_{\mathbf{v}=\mathbf{0}}, \mathbf{u} \rangle_e$$

and we have (12). $\qquad\square$

### B.4. Proof of Proposition A.2

**Proposition B.2** (Trivialized score)**.** *Let $G$ be a finite-dimensional Lie group. Under any choice of a left-invariant metric, for any positive and smooth function $p : G \to \mathbb{R}$, it holds that $\nabla_{\mathfrak{g}} p = p \nabla_{\mathfrak{g}} \log p$.*

*Proof.* For any smooth function $f : G \to \mathbb{R}$, we can turn its differential $\mathrm{d}f_g$ evaluated at $g$ into a linear functional on $\mathfrak{g}$ by pre-composing with $\mathrm{d}(L_{g^{-1}})_g^{-1}$. Concretely, for any $g \in G$ and $\mathbf{u} \in \mathfrak{g}$,

$$\langle \nabla_{\mathfrak{g}} f(g), \mathbf{u} \rangle_{\mathfrak{g}} = \mathrm{d}f_g(\mathrm{d}(L_{g^{-1}})_g^{-1} \mathbf{u}) = \left. \frac{\mathrm{d}}{\mathrm{d}t} f(g \exp(t\mathbf{u})) \right|_{t=0}$$

This is exactly the directional derivative of $f$ along the left-invariant vector field generated by $\mathbf{u}$.

We can apply the above equation to the left-trivialized score since we assumed the smoothness of each $p$. With $f = \log p$, we get

$$(\mathrm{d}(L_{g^{-1}})_g \nabla \log p(g))(\mathbf{u}) = \left. \frac{\mathrm{d}}{\mathrm{d}t} \log p(g \exp(t\mathbf{u})) \right|_{t=0}$$

By the chain rule on $\mathbb{R}$,

$$\frac{\mathrm{d}}{\mathrm{d}t} \log p(g \exp(t\mathbf{u})) = \frac{1}{p(g \exp(t\mathbf{u}))} \frac{\mathrm{d}}{\mathrm{d}t} p(g \exp(t\mathbf{u}))$$

Evaluating at $t = 0$ gives

$$\langle \nabla_{\mathfrak{g}} \log p(g), \mathbf{u} \rangle_{\mathfrak{g}} = \left. \frac{1}{p(g)} \frac{\mathrm{d}}{\mathrm{d}t} p(g \exp(t\mathbf{u})) \right|_{t=0} = \frac{\langle \nabla_{\mathfrak{g}} p(g), \mathbf{u} \rangle_{\mathfrak{g}}}{p(g)}$$

Since this holds for arbitrary $\mathbf{u} \in \mathfrak{g}$, we get $\nabla_{\mathfrak{g}} p(g) = p(g) \nabla_{\mathfrak{g}} \log p(g)$ for all $g \in G$. $\qquad\square$

## B.5. Proof of Proposition 4.1

The proof is inspired by the generator-based approach of Zhu et al. (2025); Holderrieth et al. (2024).

For an $n$-dimensional Lie group $G$ with any choice of orthonormal basis $\{\mathbf{e}_1, ..., \mathbf{e}_n\}$ of Lie algebra $\mathfrak{g}$, we recall the SDE given in (10), with Brownian motion in the Lie algebra $\mathrm{d}\mathbf{w}_t^{\mathfrak{g}} = \sum_i \mathbf{e}_i \mathrm{d}w_t^i$, drift coefficient $\phi(\cdot, t) : G \to \mathfrak{g}$, and diffusion coefficient $\gamma(t) \in \mathbb{R}$:

$$\mathrm{d}g_t = \mathrm{d}(L_{g_t})_e \left[ \phi(g_t, t) \, \mathrm{d}t + \gamma(t) \, \mathrm{d}\mathbf{w}_t^{\mathfrak{g}} \right].$$

We show some useful lemmas related to the SDE. We first derive its infinitesimal generator $\mathcal{L}_t$, which is a linear operator defined for every smooth function $f : G \to \mathbb{R}$ as the following for each $g \in G$:

$$(\mathcal{L}_t f)(g) \equiv \lim_{h \to 0^+} \frac{\mathbb{E}[f(g_{t+h}) - f(g_t)|g_t = g]}{h}.$$

**Lemma B.3.** *Assume the SDE in (10) has a smooth drift. Then its infinitesimal generator $\mathcal{L}_t$ satisfies the following for any smooth $f : G \to \mathbb{R}$, where $\Delta$ is the Laplace-Beltrami operator:*

$$\mathcal{L}_t f = \langle \nabla_{\mathfrak{g}} f, \phi(\cdot, t) \rangle_{\mathfrak{g}} + \frac{\gamma(t)^2}{2} \Delta f. \tag{13}$$

*Proof.* Let us denote $E_i \equiv \mathrm{d}(L_{[\cdot]})_e \mathbf{e}_i$ and rewrite (10) as follows

$$\mathrm{d}g_t = \Phi(g_t, t) \, \mathrm{d}t + \sum_{i=1}^n \Gamma_i(g_t, t) \, \mathrm{d}w_t^i, \quad \Phi(g, t) \equiv \mathrm{d}(L_g)_e \phi(g, t), \quad \Gamma_i(g, t) \equiv \gamma(t) E_i$$

Since $\Phi(\cdot, t), \Gamma_1(\cdot, t), ..., \Gamma_n(\cdot, t)$ are smooth vector fields on $G$, this expression makes it explicit that (10) is an Itô SDE taking $G$ as the state space. Then, by applying Lee & Chirikjian (2025, Thm. 2 and Prop. 1)[2] we get

$$\begin{aligned}
\mathcal{L}_t f &= \mathrm{d}f_{[\cdot]}(\Phi(\cdot, t)) + \frac{1}{2} \sum_{i=1}^n \mathrm{Hess}_f(\Gamma_i(\cdot, t), \Gamma_i(\cdot, t)) \\
&= \mathrm{d}f_{[\cdot]}(\Phi(\cdot, t)) + \frac{\gamma(t)^2}{2} \sum_{i=1}^n \mathrm{Hess}_f(E_i, E_i) \\
&= \mathrm{d}f_{[\cdot]}(\Phi(\cdot, t)) + \frac{\gamma(t)^2}{2} \Delta f \\
&= \langle \nabla f, \Phi(\cdot, t) \rangle + \frac{\gamma(t)^2}{2} \Delta f
\end{aligned}$$

where we use bilinearity of Hessian for the second equality and use (9) for the last equality. With left-invariance of the metric we obtain (13). $\qquad \square$

We now derive the adjoint operator $\mathcal{L}_t^*$ of the infinitesimal generator, which is defined through the following relationship for any smooth, compactly supported test function $f : G \to \mathbb{R}$ and density $\rho$ with respect to the Haar measure $\mu$ (Holderrieth et al., 2024, §A.3):

$$\int_G (\mathcal{L}_t f) \, \rho \, \mathrm{d}\mu = \int_G f \, (\mathcal{L}_t^* \rho) \, \mathrm{d}\mu. \tag{14}$$

**Lemma B.4.** *Assume the SDE in (10) has a smooth drift. The adjoint $\mathcal{L}_t^*$ of its infinitesimal generator satisfies the following for any bounded and smooth density $\rho$, where* div *is the divergence:*

$$\mathcal{L}_t^* \rho = -\mathrm{div}(\rho \, \Phi(\cdot, t)) + \frac{\gamma(t)^2}{2} \Delta \rho. \tag{15}$$

---

[2]Here we consider an extension of the results in Lee & Chirikjian (2025) to time-dependent drift and diffusion coefficients. Such an extension can be obtained since the results are based on applying the chain rule to a standard form of Stratonovich SDE with respect to only the state variable and hence we can regard $t$ as a constant.

*Proof.* From (14), Lem. B.3, and bilinearity of the metric, we get

$$\int_G f\left(\mathcal{L}_t^* \rho\right) \mathrm{d}\mu = \int_G \langle \nabla f, \rho\,\Phi(\cdot, t)\rangle \,\mathrm{d}\mu + \frac{\gamma(t)^2}{2} \int_G \rho\,\Delta f \,\mathrm{d}\mu \tag{16}$$

For the first term, we use the following relationship

$$0 = \int_G \mathrm{div}(f\,\rho\,\Phi(\cdot, t))\,\mathrm{d}\mu$$
$$= \int_G f\,\mathrm{div}(\rho\,\Phi(\cdot, t))\,\mathrm{d}\mu + \int_G \langle \nabla f, \rho\,\Phi(\cdot, t)\rangle\,\mathrm{d}\mu \tag{17}$$

where the first equality follows from the divergence theorem for the compactly supported vector field $f\,\rho\,\Phi(\cdot, t)$, and the second equality follows from the identity $\mathrm{div}(f\,X) = f\,\mathrm{div}X + \langle \nabla f, X\rangle$ with the vector field $X = \rho\,\Phi(\cdot, t)$ (Chavel, 2006, Theorem III.7.3 and p.150).

For the second term, we use the fact that $\Delta$ is self-adjoint (Chavel, 2006, Theorem III.7.4), that is

$$\int_G \rho\,\Delta f \,\mathrm{d}\mu = \int_G f\,\Delta\rho \,\mathrm{d}\mu \tag{18}$$

Plugging (17) and (18) into (16), we get

$$\int_G f\left(\mathcal{L}_t^* \rho\right) \mathrm{d}\mu = -\int_G f\,\mathrm{div}(\rho\,\Phi(\cdot, t))\,\mathrm{d}\mu + \frac{\gamma(t)^2}{2}\int_G f\,\Delta\rho\,\mathrm{d}\mu$$

Since this holds for every test function $f$, we get (15). $\qquad\square$

We are now ready to prove Prop. 4.1. We recall the forward trivialized SDE in (5):

$$\mathrm{d}g_t = \mathrm{d}(L_{g_t})_e \left[\gamma(t)\,\mathrm{d}\mathbf{w}_t^{\mathfrak{g}}\right].$$

The proof uses the adjoint Kolmogorov forward equation that describes density evolution $\rho_t$ of a stochastic process using the adjoint of its infinitesimal generator (Zhu et al., 2025; Holderrieth et al., 2024):

$$\frac{\partial}{\partial t}\rho_t = \mathcal{L}_t^* \rho_t.$$

**Proposition 4.1** (Reverse trivialized SDE). *If each $p_t(g_t)$ is smooth and positive with respect to the Haar measure, then the time-reversal of (5) is*

$$\mathrm{d}g_t = \mathrm{d}(L_{g_t})_e\left[-\gamma(t)^2\,\nabla_{\mathfrak{g}}\log p_t(g_t)\,\mathrm{d}t + \gamma(t)\,\mathrm{d}\bar{\mathbf{w}}_t^{\mathfrak{g}}\right], \tag{6}$$

*where $\bar{\mathbf{w}}_t^{\mathfrak{g}}$ is Brownian motion in $\mathfrak{g}$ run backwards in time, and $\nabla_{\mathfrak{g}}\log p_t(g_t)$ is the trivialized score, i.e., the score expressed as an element of the Lie algebra.*

*Proof.* Denoting the density of the forward process by $p_t$ and of the reverse by $\bar{p}_t$, we prove $p_t = \bar{p}_{1-t}$ for all $t \in [0, 1]$ given the initial condition $p_1 = \bar{p}_0$.

From Lem. B.4, the adjoint Kolmogorov forward equation of the forward process is

$$\frac{\partial}{\partial t}p_t = \frac{\gamma(t)^2}{2}\Delta p_t$$

Likewise, the adjoint Kolmogorov forward equation of the proposed reverse process is

$$\frac{\partial}{\partial t}\bar{p}_t = -\gamma(1-t)^2\mathrm{div}(\bar{p}_t\,\nabla\log\bar{p}_t) + \frac{\gamma(1-t)^2}{2}\Delta\bar{p}_t = -\frac{\gamma(1-t)^2}{2}\Delta\bar{p}_t$$

where we used $\mathrm{div}(\bar{p}_t\,\nabla\log\bar{p}_t) = \mathrm{div}(\nabla\bar{p}_t) = \Delta\bar{p}_t$ for the second equality.

Then, the partial derivative of $\bar{p}_{1-t}$ with respect to $t$ is

$$\frac{\partial}{\partial t}\bar{p}_{1-t} = \frac{\gamma(t)^2}{2}\Delta\bar{p}_{1-t}$$

Since this precisely matches the partial derivative of $p_t$ with respect to $t$, given the initial condition $p_1 = \bar{p}_0$ we have that $p_t = \bar{p}_{1-t}$ for all $t \in [0, 1]$, completing the proof. $\qquad\square$

### B.6. Proof of Proposition 4.2

Before proving our main score identity, we state two lemmas regarding conditional random variables on general Lie groups. We distinguish between random variables and their realizations for clarity.

**Lemma B.5.** *Let $G$ be a finite-dimensional Lie group, and let $X, W$ be independent random variables with densities $p_X, p_W$ with respect to the left-invariant measure $\mu$. Let $Y \equiv XW$. Then we have:*

$$p_{Y|X}(y|x) = p_W(x^{-1}y). \tag{19}$$

*Proof.* For any measurable set $A \subseteq G$, we have

$$\mathbb{P}\left(Y \in A | X = x\right) = \mathbb{P}\left(xW \in A\right) = \mathbb{P}\left(W \in x^{-1}A\right) = \int_{x^{-1}A} p_W(w)\,\mathrm{d}\mu(w)$$

where the first equality is due to independence of $X$ and $W$.

Let us apply change of variable $y \equiv xw$. Since $\mathrm{d}\mu(w) = \mathrm{d}\mu(x^{-1}y) = \mathrm{d}\mu(y)$, we have

$$\mathbb{P}\left(Y \in A | X = x\right) = \int_{x^{-1}A} p_W(w)\,\mathrm{d}\mu(w) = \int_A p_W(x^{-1}y)\,\mathrm{d}\mu(y)$$

Since this holds for every measurable set $A \subseteq G$, we get (19). $\qquad\square$

For the next lemma, we introduce the concept of the modular function $\lambda : G \to \mathbb{R}_{>0}$ of a Lie group $G$ that specifies how the left-invariant measure $\mu$ changes under right multiplication. Specifically, for every measurable set $A \subseteq G$ and every $g \in G$, it holds that $\mu(Ag) = \lambda(g)\mu(A)$, and in particular, we have that $\mathrm{d}\mu(g^{-1}) = \lambda(g^{-1})\,\mathrm{d}\mu(g) = \lambda(g)^{-1}\,\mathrm{d}\mu(g)$ (Folland, 2015, §2.4).

**Lemma B.6.** *Let $G$ be a finite-dimensional Lie group, and let $X, W$ be independent random variables with densities $p_X, p_W$ with respect to $\mu$. Let $Y \equiv XW$. Then we have, with $\lambda$ the modular function:*

$$p_Y(y) = \int_G p_W(w)\,p_X(yw^{-1})\,\lambda(w)^{-1}\,\mathrm{d}\mu(w). \tag{20}$$

*Proof.* From Lem. B.5, we obtain

$$p_Y(y) = \int_G p_X(x)\,p_{Y|X}(y|x)\,\mathrm{d}\mu(x) = \int_G p_X(x)\,p_W(x^{-1}y)\,\mathrm{d}\mu(x)$$

By applying change of variable $w = x^{-1}y$ with $\mathrm{d}\mu(x) = \mathrm{d}\mu(yw^{-1}) = \mathrm{d}\mu(w^{-1}) = \lambda(w)^{-1}\mathrm{d}\mu(w)$, we get (20). $\qquad\square$

We are now ready to prove the main score identity.

**Proposition 4.2** (Trivialized target score identity)**.** *For the forward SDE in (5), we have*

$$\nabla_{\mathfrak{g}} \log p_t(g_t) = \int_G \nabla_{\mathfrak{g}} \log p_0(g_0)\,p_{0|t}(g_0 \mid g_t)\mathrm{d}\mu(g_0) \tag{7}$$

*where the argument of $\log p_0$ is interpreted as $g_t b$ for $b \equiv g_t^{-1} g_0$, and $\nabla_{\mathfrak{g}}$ is taken with respect to $g_t$.*

*Proof.* We note that $g_0$ and $g_t$ as random variables satisfy a relationship $g_t = g_0 w_t$ where $g_0 \sim p_0$ and $w_t \sim k_t$ are independent. By applying Lem. B.6, we get

$$p_t(g_t) = \int_G k_t(w_t)\,p_0(g_t w_t^{-1})\,\lambda(w_t)^{-1}\,\mathrm{d}\mu(w_t) \tag{21}$$

By taking trivialized gradient $\nabla_{\mathfrak{g}}$ with respect to the variable $g_t$ and noting that it is a linear operator,

$$\nabla_{\mathfrak{g}} p_t(g_t) = \int_G k_t(w_t)\,\nabla_{\mathfrak{g}} p_0(g_t w_t^{-1})\,\lambda(w_t)^{-1}\,\mathrm{d}\mu(w_t) \tag{22}$$

Then, using the identity $\nabla_{\mathfrak{g}} p = p \nabla_{\mathfrak{g}} \log p$ from Prop. A.2 we get

$$\nabla_{\mathfrak{g}} \log p_t(g_t) = \int_G \nabla_{\mathfrak{g}} \log p_0(g_t w_t^{-1}) \frac{k_t(w_t)\, p_0(g_t w_t^{-1})}{p_t(g_t)} \lambda(w_t)^{-1} \, \mathrm{d}\mu(w_t)$$

By change of variable $g_0 = g_t w_t^{-1}$ with $\lambda(w_t)^{-1} \, \mathrm{d}\mu(w_t) = \mathrm{d}\mu(w_t^{-1}) = \mathrm{d}\mu(g_t w_t^{-1}) = \mathrm{d}\mu(g_0)$, and using the relationship $p_{t|0}(g_t|g_0) = k_t(g_0^{-1} g_t) = k_t(w_t)$ obtained by Lem. B.5, we get

$$\nabla_{\mathfrak{g}} \log p_t(g_t) = \int_G \nabla_{\mathfrak{g}} \log p_0(g_0) \frac{p_{t|0}(g_t|g_0)\, p_0(g_0)}{p_t(g_t)} \, \mathrm{d}\mu(g_0)$$

where we remark that $\nabla_{\mathfrak{g}}$ is taken with respect to $g_t$, so $\nabla_{\mathfrak{g}} \log p_0(g_0)$ is understood as $\nabla_{\mathfrak{g}} \log p_0(g_t b)$ for $b$ having the value of $g_t^{-1} g_0$. Using Bayes' rule we get (7). $\qquad \square$

### B.7. Proof of Proposition 4.3

The proof is inspired by the convolution argument of Akhound-Sadegh et al. (2024), or equivalently an importance sampling estimation applied to target score identity (De Bortoli et al., 2024).

**Proposition 4.3** (Monte Carlo score estimator). *For the forward SDE in (5), where $p_0$ is a Boltzmann density specified by a smooth energy $E$, we have*

$$\nabla_{\mathfrak{g}} \log p_t(g_t) = \nabla_{\mathfrak{g}} \log \int_G k_t(w)\, h(g_t, w) \, \mathrm{d}\mu(w),$$

$$h(g_t, w) = \exp\big(-E(g_t w^{-1}) - \log \lambda(w)\big)$$

$\qquad (8)$

*where $\lambda$ accounts for the change of the Haar measure under inversion, $\mathrm{d}\mu(w^{-1}) = \lambda(w)^{-1} \mathrm{d}\mu(w)$.*

*Proof.* We start at (22) and use $\nabla_{\mathfrak{g}} p_t = p_t \nabla_{\mathfrak{g}} \log p_t$ with (21) to get

$$\nabla_{\mathfrak{g}} \log p_t(g) = \frac{\int_G k_t(w)\, \nabla_{\mathfrak{g}} p_0(g w^{-1})\, \lambda(w)^{-1} \, \mathrm{d}\mu(w)}{\int_G k_t(w)\, p_0(g w^{-1})\, \lambda(w)^{-1} \, \mathrm{d}\mu(w)}$$

Then using $p_0(\cdot) = e^{-E(\cdot)}/Z$ with the normalization constant $Z$ canceling out, we get

$$\nabla_{\mathfrak{g}} \log p_t(g) = \frac{\int_G k_t(w)\, \nabla_{\mathfrak{g}} e^{-E(g_t w^{-1}) - \log \lambda(w)} \, \mathrm{d}\mu(w)}{\int_G k_t(w)\, e^{-E(g_t w^{-1}) - \log \lambda(w)} \, \mathrm{d}\mu(w)}$$

Using linearity to move $\nabla_{\mathfrak{g}}$ with respect to $g$ out of the integration, and then applying the identity $\nabla_{\mathfrak{g}} f = f \nabla_{\mathfrak{g}} \log f$ with $f$ the denominator, we get (8) (by writing $g_t$ instead of $g$). $\qquad \square$

### B.8. Monte Carlo estimation and consistency

We discuss sample-based estimation of the trivialized score based on Prop. 4.3 and prove its consistency. Starting from (8) and using Monte Carlo estimation for the integral with $N$ samples, we get:

$$\nabla_{\mathfrak{g}} \log p_t(g) \approx \nabla_{\mathfrak{g}} \log \frac{1}{N} \sum_{i=1}^{N} \exp\big(-E(g w^{(i)-1}) - \log \lambda(w^{(i)})\big), \quad w^{(i)} \sim k_t. \qquad (23)$$

The $1/N$ factor does not affect the gradient, and hence can be removed. We then obtain an expression of the form $\log \sum_i \exp$ inside the gradient, which allows for a numerically stable evaluation using the `logsumexp` trick (Akhound-Sadegh et al., 2024).

We note that the estimator is not unbiased because it has sample mean inside the concave $\log$ function. The bias can be understood as the Jensen gap. Nevertheless, we show that it is a consistent estimator.

**Proposition B.7.** *For the forward SDE in (5) where $p_0$ is a Boltzmann density specified by a smooth energy $E$, under proper integrability and positivity conditions, (23) is a consistent estimator of $\nabla_{\mathfrak{g}} \log p_t$.*

*Proof.* For each $g \in G$, let us denote $h(g, w) \equiv e^{-E(gw^{-1}) - \log \lambda(w)}$ and define

$$J(g) \equiv \int_G k_t(w) \nabla_{\mathfrak{g}} h(g, w) \, \mathrm{d}\mu(w), \quad Z(g) \equiv \int_G k_t(w) \, h(g, w) \, \mathrm{d}\mu(w)$$

and define the corresponding empirical means

$$\hat{J}_N(g) \equiv \frac{1}{N} \sum_{i=1}^{N} \nabla_{\mathfrak{g}} h(g, w^{(i)}), \quad \hat{Z}_N(g) \equiv \frac{1}{N} \sum_{i=1}^{N} h(g, w^{(i)}), \quad w^{(i)} \sim k_t$$

Then (8) and (23) can be written respectively as

$$\nabla_{\mathfrak{g}} \log p_t(g) = \nabla_{\mathfrak{g}} \log Z(g) = \frac{J(g)}{Z(g)}, \quad \nabla_{\mathfrak{g}} \log \hat{Z}_N(g) = \frac{\hat{J}_N(g)}{\hat{Z}_N(g)}$$

Assume integrability and positivity conditions

$$\int_G k_t(w) \, \|\nabla_{\mathfrak{g}} h(g, w)\| \, \mathrm{d}\mu(w) < \infty, \quad 0 < \int_G k_t(w) \, h(g, w) \, \mathrm{d}\mu(w) < \infty$$

By the strong law of large numbers

$$\hat{J}_N(g) \xrightarrow{\text{a.s.}} J(g), \quad \hat{Z}_N(g) \xrightarrow{\text{a.s.}} Z(g)$$

Then, by continuous mapping theorem (Shao, 2003, Theorem 1.10 and Example 1.30)

$$\frac{\hat{J}_N(g)}{\hat{Z}_N(g)} \xrightarrow{\text{a.s.}} \frac{J(g)}{Z(g)}$$

thus we have $\nabla_{\mathfrak{g}} \log \hat{Z}_N(g) \xrightarrow{\text{a.s.}} \nabla_{\mathfrak{g}} \log Z(g) = \nabla_{\mathfrak{g}} \log p_t(g)$. □

## B.9. Implementation details for diffusion sampling

We discuss numerical computation of diffusion sampling $\hat{g}_0 \sim p_0 \propto e^{-E(g)}$ given $E : G \to \mathbb{R}$. The sampling follows an Euler time-discretization (11) of the reverse diffusion (6). For step size $\Delta t > 0$ we use the following for decreasing step indices $m = M, ..., 1$ with $M = 1/\Delta t$:

$$\hat{g}_{m-1} = \hat{g}_m \exp\left(\gamma(m\Delta t)^2 \nabla_{\mathfrak{g}} \log p_{m\Delta t}(\hat{g}_m) \, \Delta t + \gamma(m\Delta t) \, \bar{\mathbf{z}}_m\right), \quad \bar{\mathbf{z}}_m \sim \mathcal{N}(\mathbf{0}, \Delta t \mathbf{I}),$$

$$\hat{g}_M \overset{d}{=} \hat{w}_M \sim k_1,$$

where we approximate $k_1 \approx p_1$ for initialization as usually done for variance-exploding diffusion (Song et al., 2020). To run the sampling, we first need to sample from $k_1$ and at each step compute the MC estimator of the score $\nabla_{\mathfrak{g}} \log p_t$ (23). This translates to the following requirements:

1. A method to sample $w \sim k_t$,

2. A method to calculate $\log \lambda(w)$,

3. A method to compute trivialized gradient $\nabla_{\mathfrak{g}}$ of general $f : G \to \mathbb{R}$.

For the first item, we recall the relationship $g_t = g_0 w_t$ with $g_0 \sim p_0$ and $w_t \sim k_t$ independent (proof of Prop. 4.2), and recall the forward process (5). Together, these imply that $w_t \sim k_t$ is described by the following SDE which is identical to the forward SDE but starts at the identity:

$$\mathrm{d}w_t = \mathrm{d}(L_{w_t})_e \left[\gamma(t) \, \mathrm{d}\mathbf{w}_t^{\mathfrak{g}}\right], \quad w_0 = e.$$

This suggests a natural Euler time-discretization for sampling. For step indices $m = 1, ..., M$ we use:

$$\hat{w}_{m+1} = \hat{w}_m \exp\left(\gamma(m\Delta t)\,\mathbf{z}_m\right), \quad \mathbf{z}_m \sim \mathcal{N}(\mathbf{0}, \Delta t\mathbf{I}), \quad \hat{w}_0 = e, \tag{24}$$

and treat $\hat{w}_m \sim k_{m\Delta t}$.

For the second item, to evaluate the modular function $\lambda(\hat{w}_m)$ for $\hat{w}_m \sim k_{m\Delta t}$ we use the fact that $\hat{w}_m$ can always be written as the following, thanks to the structure of time-discretization (24):

$$\hat{w}_m = \exp(\mathbf{v}_1)\exp(\mathbf{v}_2)\cdots\exp(\mathbf{v}_{m-1}),$$

where each $\mathbf{v}_i = \gamma(i\Delta t)\mathbf{z}_i$ is a Lie algebra element obtained at each sampling step. This allows us to evaluate the modular function only using the knowledge of the Lie bracket $[\cdot, \cdot] : \mathfrak{g} \times \mathfrak{g} \to \mathfrak{g}$:

**Proposition B.8.** *Let $G$ be a connected $n$-dimensional Lie group. Under a choice of orthonormal basis $\{\mathbf{e}_1, ..., \mathbf{e}_n\}$, with structure constants $c_{ij}^k$ satisfying $[\mathbf{e}_i, \mathbf{e}_j] = \sum_{k=1}^n c_{ij}^k \mathbf{e}_k$, let us denote:*

$$\mathbf{a} \equiv \begin{bmatrix} \sum_{j=1}^n c_{1j}^j \\ \cdots \\ \sum_{j=1}^n c_{nj}^j \end{bmatrix} \in \mathbb{R}^n, \quad \mathbf{b}_1 \equiv \begin{bmatrix} \langle \mathbf{e}_1, \mathbf{v}_1 \rangle_{\mathfrak{g}} \\ \cdots \\ \langle \mathbf{e}_n, \mathbf{v}_1 \rangle_{\mathfrak{g}} \end{bmatrix} \quad \cdots \quad \mathbf{b}_{m-1} \equiv \begin{bmatrix} \langle \mathbf{e}_1, \mathbf{v}_{m-1} \rangle_{\mathfrak{g}} \\ \cdots \\ \langle \mathbf{e}_n, \mathbf{v}_{m-1} \rangle_{\mathfrak{g}} \end{bmatrix} \in \mathbb{R}^n.$$

*Then the following holds:*

$$\lambda(\hat{w}_m) = e^{-\mathbf{a}^\top \mathbf{b}_1 - \mathbf{a}^\top \mathbf{b}_2 - ... - \mathbf{a}^\top \mathbf{b}_{m-1}}. \tag{25}$$

*Proof.* For every connected Lie group $G$, we have (Folland, 2015, Prop. 2.30)

$$\lambda(g) = \det \mathrm{Ad}(g^{-1})$$

where $\mathrm{Ad}(\cdot) : \mathfrak{g} \to \mathfrak{g}$ is the adjoint action of $G$ on the Lie algebra. Using its properties as a linear group representation (Kirillov, 2008, Example 4.8), we get

$$\lambda(\hat{w}_m) = \frac{1}{\det \mathrm{Ad}(\hat{w}_m)} = \frac{1}{\det \mathrm{Ad}(\exp(\mathbf{v}_1)\cdots\exp(\mathbf{v}_{m-1}))}$$

$$= \frac{1}{\det \mathrm{Ad}(\exp(\mathbf{v}_1))} \times \cdots \times \frac{1}{\det \mathrm{Ad}(\exp(\mathbf{v}_{m-1}))}$$

Then, using that $\det \mathrm{Ad}(\exp(\mathbf{v})) = \det e^{\mathrm{ad}(\mathbf{v})} = e^{\mathrm{tr}(\mathrm{ad}(\mathbf{v}))}$, where $\mathrm{ad}(\mathbf{v}) : \mathbf{u} \mapsto [\mathbf{v}, \mathbf{u}]$ is a linear map on $\mathfrak{g}$ (Kirillov, 2008, Lem. 3.14) with properties following from bilinearity of the Lie bracket

$$\mathrm{ad}(\mathbf{v})(\mathbf{e}_i) = [\mathbf{v}, \mathbf{e}_i] = \sum_j \langle \mathbf{e}_j, \mathbf{v} \rangle_{\mathfrak{g}} [\mathbf{e}_j, \mathbf{e}_i] = \sum_j \langle \mathbf{e}_j, \mathbf{v} \rangle_{\mathfrak{g}} \sum_k c_{ji}^k \mathbf{e}_k$$

$$\mathrm{tr}(\mathrm{ad}(\mathbf{v})) = \sum_i \langle \mathbf{e}_i, \mathrm{ad}(\mathbf{v})(\mathbf{e}_i) \rangle_{\mathfrak{g}} = \sum_i \langle \mathbf{e}_i, \mathbf{v} \rangle_{\mathfrak{g}} \sum_j c_{ij}^j$$

we obtain (25). □

Finally, for the last item, we use Prop. A.1 to directly evaluate the trivialized gradient using automatic differentiation.

We finish the section with a description of the choice of noise schedule $\gamma$ and sampler configurations. We adopt the geometric noise schedule from Song et al. (2020), defined for a choice of $\gamma_{\min} < \gamma_{\max}$ as $\gamma(t) \equiv \gamma_{\min} \cdot (\gamma_{\max}/\gamma_{\min})^t$. It starts at $\gamma(0) = \gamma_{\min}$ and gradually increases to $\gamma(1) = \gamma_{\max}$. We provide the specific configurations of the TIED sampler used in the experiments in Tab. 3.

*Table 3.* Configurations of the TIED sampler used in our experiments.

| Group | Domain | Energy | Noise schedule $(\gamma_{\min}, \gamma_{\max})$ | Step size $\Delta t$ | MC sample size $N$ |
|---|---|---|---|---|---|
| SO(10) | Synthetic | Quadratic | (0.01, 10) | 1/100 | 100 |
| $\text{Aff}(2, \mathbb{R})$ | Image | VAE | (0.1, 1) | 1/50 | 4 |
| $\text{Aff}(2, \mathbb{R})$ | Image | Classifier | (0.1, 1) | 1/50 | 2 |
| $\text{PGL}(3, \mathbb{R})$ | Image | VAE | (0.05, 0.5) | 1/50 | 4 |
| $\text{PGL}(3, \mathbb{R})$ | Image | Classifier | (0.01, 0.5) | 1/50 | 2 |
| $\text{SL}(2, \mathbb{R}) \ltimes \text{H}(1, \mathbb{R})$ | PDE | Boundary | (0.01, 0.5) | 1/50 | 10 |
| $\text{SL}(2, \mathbb{R}) \ltimes (\mathbb{R}^2, +)$ | PDE | Boundary | (0.01, 0.5) | 1/50 | 10 |

*Table 4.* MNIST classification test wall-clock runtime measured on a single NVIDIA A6000 GPU.

| dataset | MNIST | |
|---|---|---|
| test transformations | $\text{Aff}(2, \mathbb{R})$ | $\text{PGL}(3, \mathbb{R})$ |
| affConv / homConv (training time) | 36 hours | 36 hours |
| Energy: VAE evidence lower bound (+ adv. reg.) | | |
| ResNet18 + ITS | 13 mins | n/a |
| ResNet18 + FoCal | 7 hours | 5 hours |
| ResNet18 + Kinetic Langevin | 90 mins | 93 mins |
| ResNet18 + LieLAC | 75 mins | 75 mins |
| ResNet18 + TIED (Ours) | 50 mins | 50 mins |
| Energy: Classifier logit confidence | | |
| ResNet18 + ITS | 12 mins | n/a |
| ResNet18 + FoCal | 7 hours | 4.5 hours |
| ResNet18 + Kinetic Langevin | 65 mins | 73 mins |
| ResNet18 + LieLAC | 30 mins | 32 mins |
| ResNet18 + TIED (Ours) | 37 mins | 36 mins |

## B.10. Other implementation details

Ensembling over augmentations generally improves the performance of pretrained neural networks (Shanmugam et al., 2021). In the context of test-time equivariance, an ensemble of the form (3) can be used (Kim et al., 2023), which performs Monte Carlo averaging of the estimator $g \cdot f \left( g^{-1} \cdot \tilde{\mathbf{x}} \right)$ with respect to $p \left( g \mid \tilde{\mathbf{x}} \right)$. We use an alternative ensemble scheme that, after diffusion sampling from $p \left( g \mid \tilde{\mathbf{x}} \right)$, selects the sample with lowest energy (or equivalently highest likelihood). This can be interpreted as sampling from a sharpened distribution (CDF raised to a power) then performing a one-sample Monte Carlo estimate. It preserves equivariance:

**Proposition B.9.** *Given a G-equivariant* $p \left( g \mid \mathbf{x} \right)$*, denote by* $p^* \left( g \mid \mathbf{x} \right)$ *the density of maximum-likelihood element among independent samples* $g_1, ..., g_K \sim p \left( \cdot \mid \mathbf{x} \right)$*, which we assume is unique almost surely. Then* $p^* \left( g \mid \mathbf{x} \right)$ *is G-equivariant.*

*Proof.* By equivariance, the samples from $p \left( \cdot \mid h \cdot \mathbf{x} \right)$ are distributed as $hg_1, ..., hg_K$ where $g_1, ..., g_K \sim p \left( \cdot \mid \mathbf{x} \right)$. Moreover, $p \left( hg_i \mid h \cdot \mathbf{x} \right) = p \left( g_i \mid \mathbf{x} \right)$ so the ordering of the likelihoods of $g_1, ..., g_K$ is unchanged. Hence, if $g^*$ is the maximizer among $g_1, ..., g_K$, then $hg^*$ is the maximizer among $hg_1, ..., hg_K$. Therefore, $g^* \mid h \cdot \mathbf{x} \overset{d}{=} h(g^* \mid \mathbf{x})$. Equivalently, in terms of densities with respect to the left Haar measure, $p^* \left( g \mid \mathbf{x} \right) = p^* \left( hg \mid h \cdot \mathbf{x} \right)$. $\square$

We use this scheme in TIED throughout, using $K = 64$, matching multiple initializations in LieLAC (Shumaylov et al., 2024). We take this cost of ensembling into account when comparing to other methods.

Additionally, when computing the energy (4) we drop the log-determinant of the Jacobian, leading to $E(g) \approx E_{\mathbf{x}}(g^{-1} \cdot \tilde{\mathbf{x}})$. This approximation makes the computation more efficient while having a negligible impact on performance.

## C. Supplementary results

We provide supplementary results that could not be included in the main text due to space constraints.

**Runtime efficiency.** In Tabs. 4 and 5, we provide the wall-clock runtime of MNIST classification and PDE solving experiments, respectively. Time cost of TIED is lower than FoCal and Langevin, thanks to parallelizability, and comparable

*Table 5.* PDE solving wall-clock runtime measured on a single NVIDIA 6000 GPU.

| PDE | 1D Heat Eq. | 1D Heat Eq. + Data Aug. | 1D Burgers Eq. |
|---|---|---|---|
| Test transformations | $\mathrm{SL}(2,\mathbb{R}) \ltimes \mathrm{H}(1,\mathbb{R})$ | $\mathrm{SL}(2,\mathbb{R}) \ltimes \mathrm{H}(1,\mathbb{R})$ | $\mathrm{SL}(2,\mathbb{R}) \ltimes (\mathbb{R}^2,+)$ |
| Energy: Distance to training domain | | | |
| DeepONet + FoCal | 13 mins | 17 mins | 7.5 mins |
| DeepONet + Kinetic Langevin | 75 mins | 75 mins | 1 hour |
| DeepONet + LieLAC | 20 secs | 20 secs | 17 secs |
| DeepONet + TIED (Ours) | 30 secs | 30 secs | 20 secs |

*Table 6.* Test accuracy of TIED across MC sample size ($N$) and sampling steps ($M$) in affNIST using the classifier confidence energy.

| | $N = 1$ | $N = 2$ | $N = 5$ | $N = 10$ |
|---|---|---|---|---|
| $M = 10$ | 75.45% | 76.50% | 77.99% | 78.36% |
| $M = 20$ | 79.62% | 80.37% | 81.23% | 81.86% |
| $M = 50$ | 82.00% | 82.63% | 83.15% | 83.41% |
| $M = 100$ | 82.88% | 83.21% | 83.33% | 83.78% |

*Table 7.* Test accuracy of TIED across noise schedules in affNIST classification using the classifier confidence energy.

| Noise schedule $(\gamma_{\min}, \gamma_{\max})$ | Accuracy |
|---|---|
| $(0.05, 1.0)$ | 83.24% |
| $(0.1, 1.0)$ | 82.63% |
| $(0.2, 1.0)$ | 81.45% |
| $(0.1, 0.5)$ | 79.74% |
| $(0.1, 2.0)$ | 77.97% |
| $(1.0, 1.0)$ | 60.00% |
| $(0.1, 0.1)$ | 59.80% |

to LieLAC, while outperforming all baselines in accuracy. TIED is slightly more costly than ITS, although we remark that ITS is specialized for affine transformations. We use 8 search levels for ITS, which we found necessary for a reasonable result, a more generous resource allocation than the original paper (Schmidt & Stober, 2024) which considered 3–5 levels. Increasing the search levels or other parameters did not lead to further gains, consistent with Schmidt & Stober (2024, Fig. 11).

**Cost-accuracy tradeoff.** While we restrict the cost of our method to be comparable with baselines, we find that its performance scales smoothly with compute. To illustrate this, we perform a study on affNIST with classifier energy (§5.2), varying the MC sample size $N$ for score estimation and the number of sampling steps $M$. The results are in Tab. 6. Even in the most resource-limited regime ($M = 10, N = 1$), it achieves 75.45% accuracy and outperforms the baselines in Tab. 1.

**Ablation analysis.** For an ablation study, we consider the diffusion schedule $\gamma(t) = \gamma_{\min} \cdot (\gamma_{\max}/\gamma_{\min})^t$. The results are in Tab. 7. Setting $\gamma(0) = \gamma_{\min}$ to be reasonably small, and $\gamma(1) = \gamma_{\max}$ to be reasonably large, offers stable results, while enforcing $\gamma(0) = \gamma(1)$ leads to degraded accuracy. This is expected from our variance-exploding formulation, which works best when it anneals between large-diffusion noise and small perturbations of clean samples (Song et al., 2020). Enforcing it to only perform large diffusion does not allow diffusion to consolidate towards the clean samples, while only allowing small perturbations makes it inaccurate as the noising kernel no longer approximates the marginal noise well (§4.2).

**Additional visualization.** Figs. 7 to 11 visualize affNIST and homNIST images transformed by our method and baselines under the classifier confidence energy.

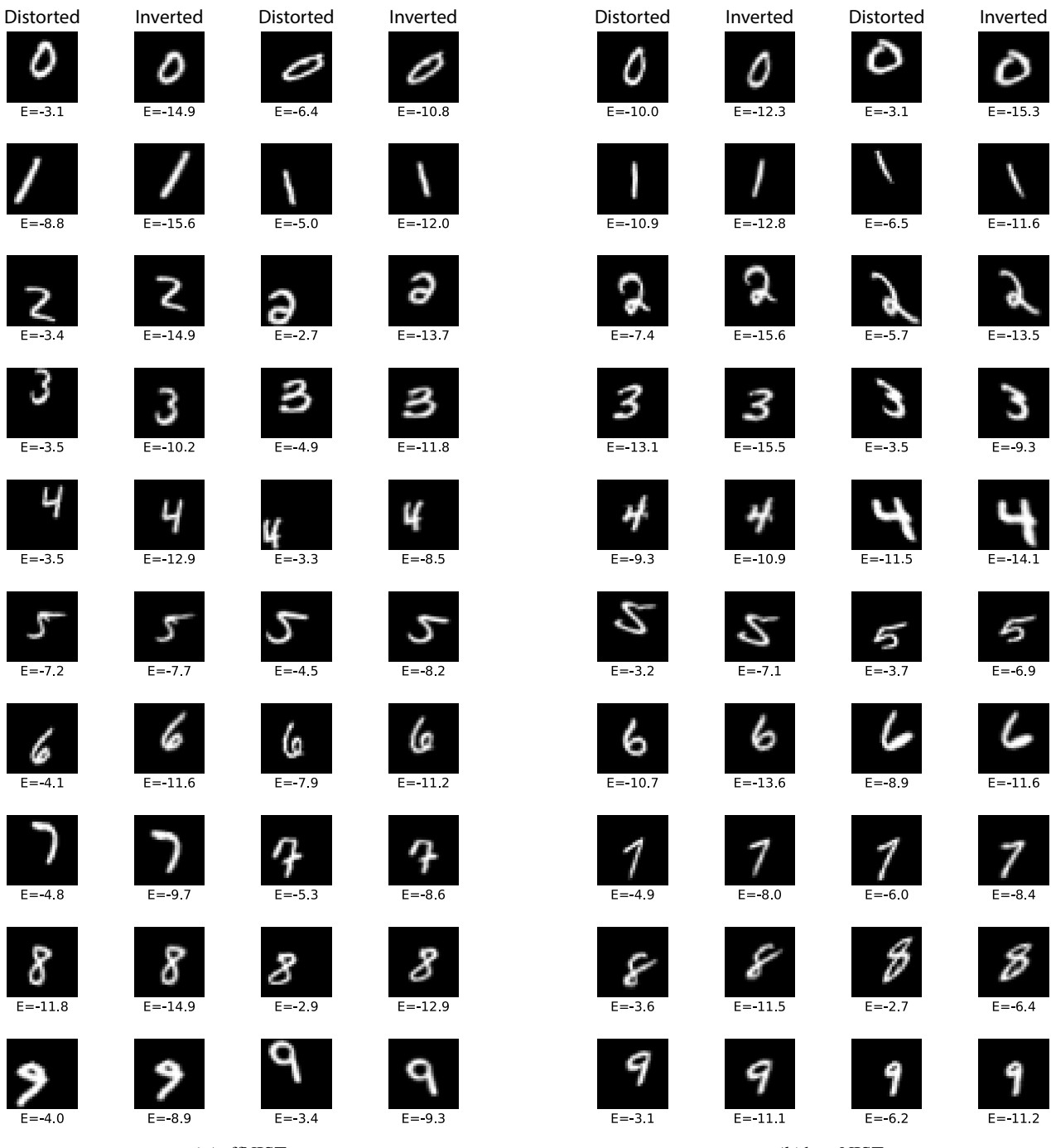

*(a)* affNIST

*(b)* homNIST

*Figure 7.* TIED examples.

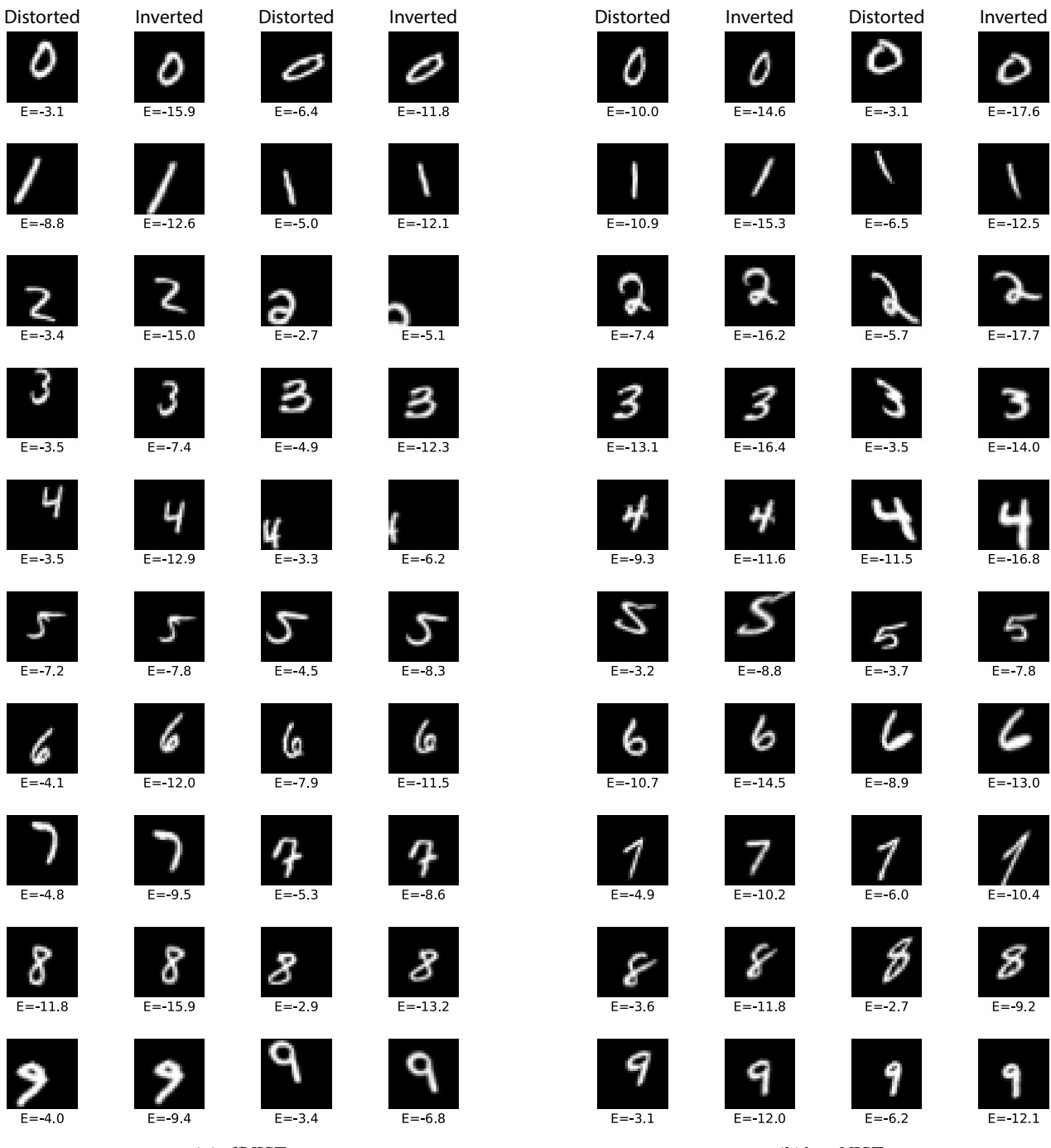

*(a)* affNIST                          *(b)* homNIST

*Figure 8.* LieLAC examples.

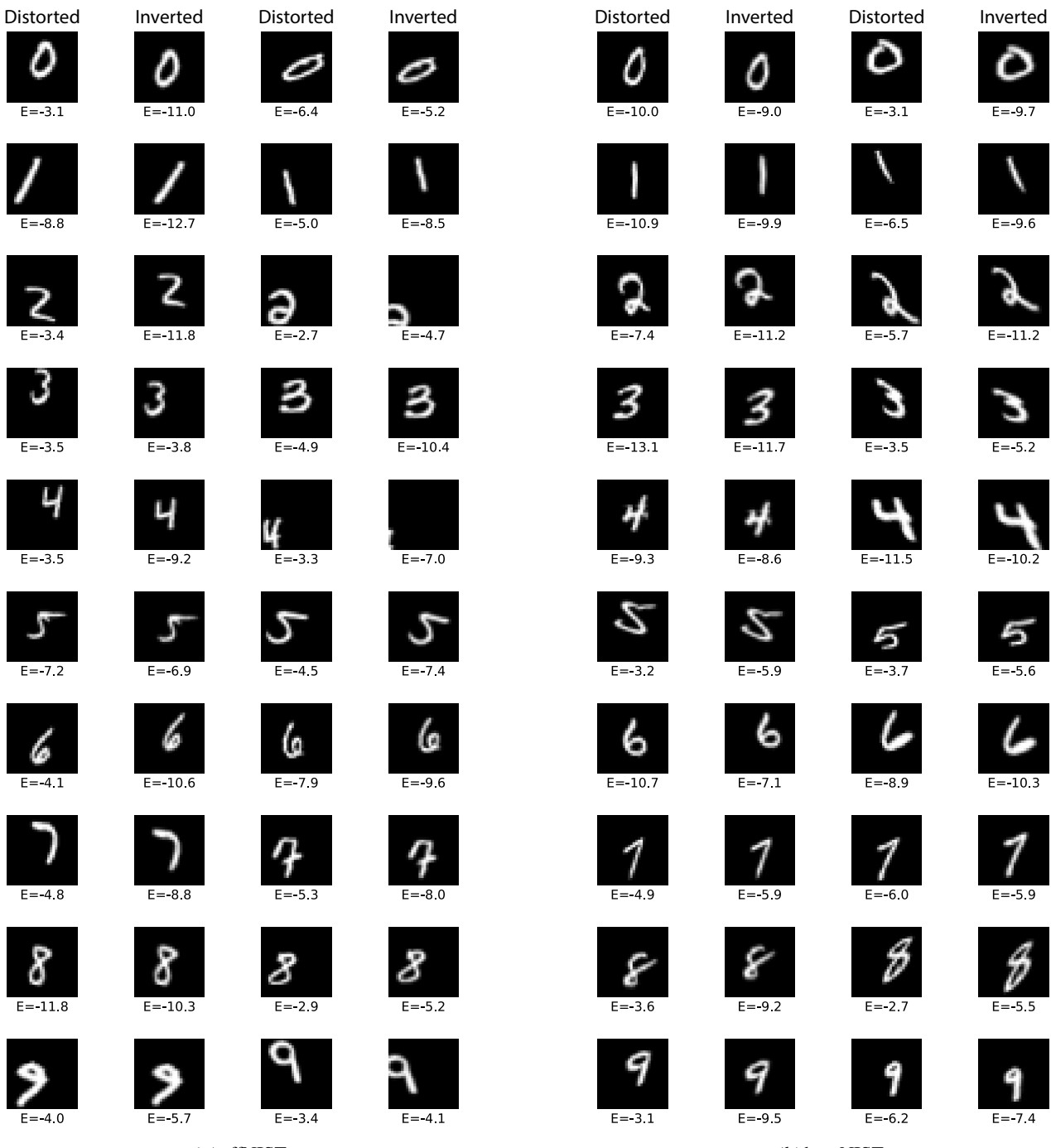

*(a)* affNIST

*(b)* homNIST

*Figure 9.* FoCal examples.

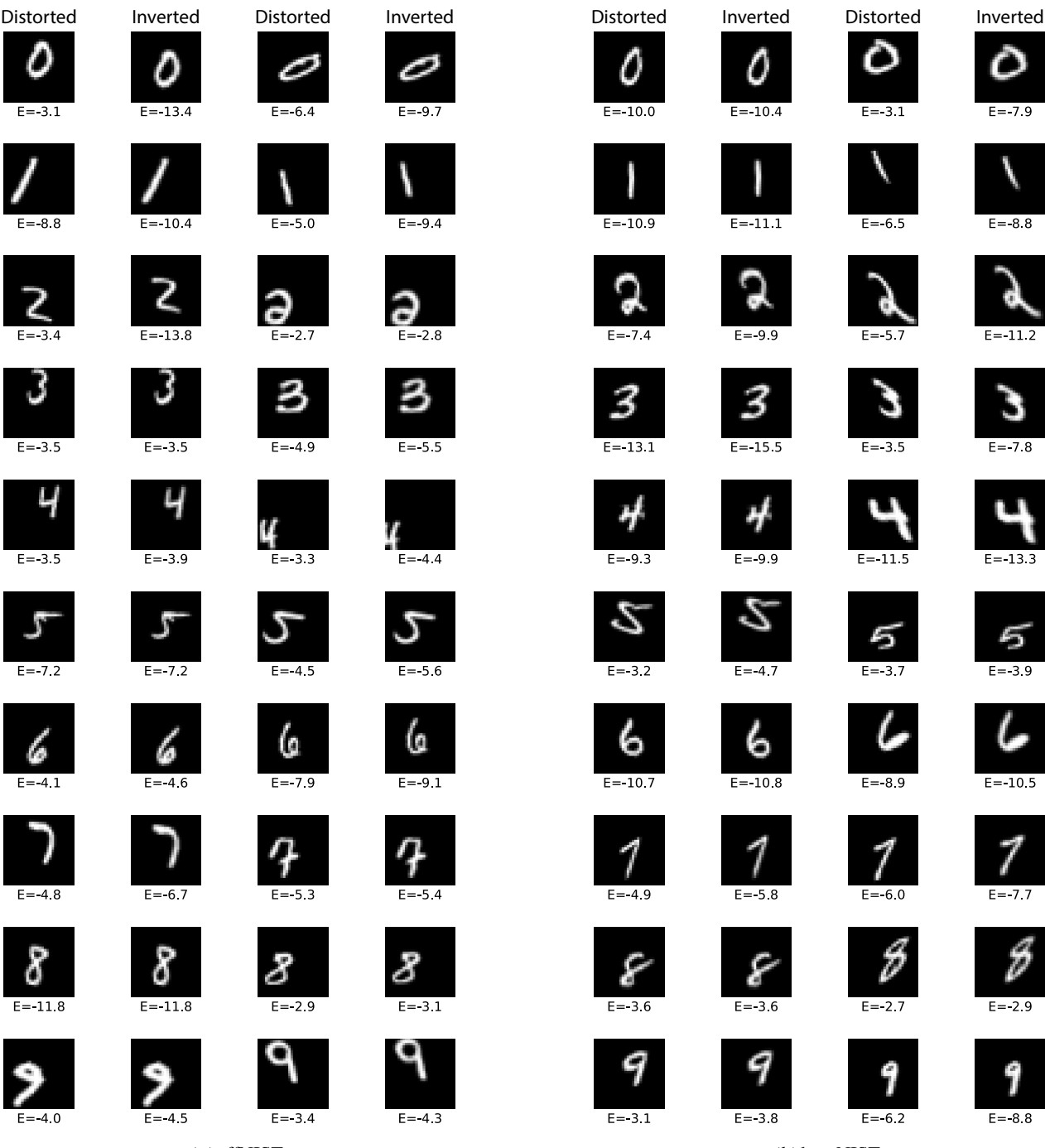

*(a)* affNIST

*(b)* homNIST

*Figure 10.* Kinetic Langevin examples.

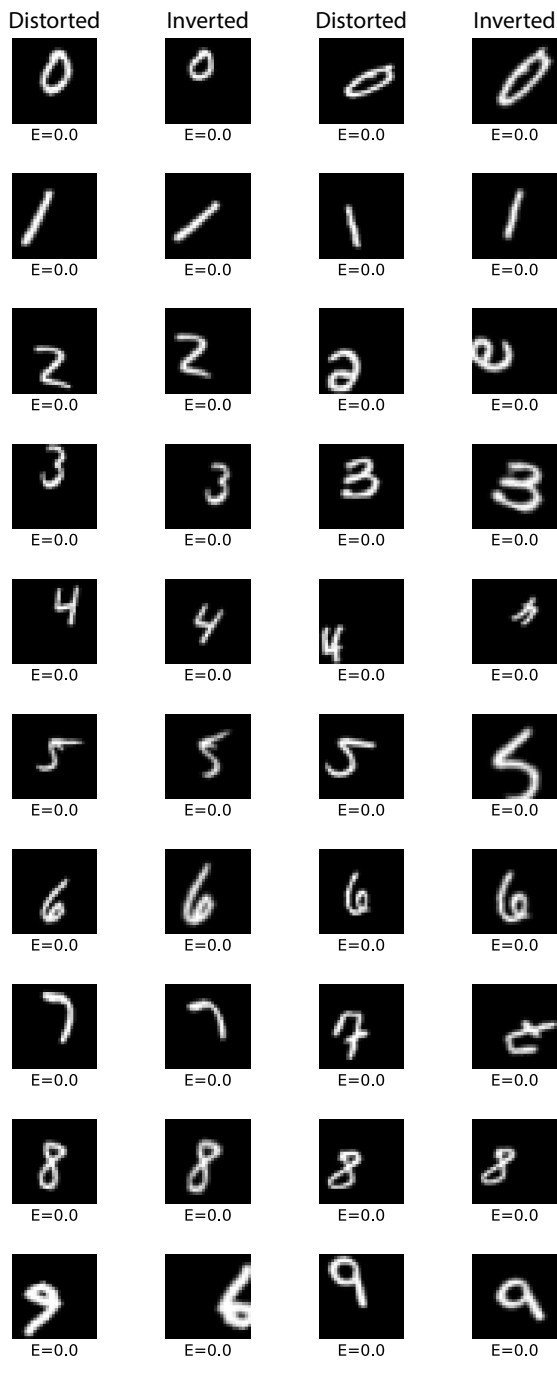

*(a)* affNIST

*Figure 11.* ITS examples.

