# OpenReview forum: "Inverting Data Transformations via Diffusion Sampling"
_ICML.cc/2026/Conference — ICML 2026 regular_

### Official Review · Reviewer_RyvQ · 2026-03-09

**Soundness:** 3
**Presentation:** 2
**Significance:** 2
**Originality:** 3
**Overall Recommendation:** 3
**Confidence:** 2

**Summary:**

This paper studies the problem of inverting unknown data transformations by sampling directly on the transformation group rather than reconstructing in data space. It develops a diffusion-based method on general Lie groups to improve the test-time equivariance and robustness of pretrained models. The paper formulates the posterior over inverse transformations as a Boltzmann distribution induced by an energy function, introduces TIED (Transformation-Inverting Energy Diffusion), and derives a target-score identity that enables reverse diffusion using only the energy function, without requiring direct samples from the transformation posterior. The method is applied to image classification under affine and homography shifts and to PDE operators under symmetry transformations, where it outperforms several optimization and sampling baselines. The derivations in the paper are hard to understand to a typical ICML reader.

**Compliance With Llm Reviewing Policy:**

Affirmed.

**Final Justification:**

This study's core idea concerns transformation inversion via diffusion on Lie groups, and this manuscript's specific area is test-time equivariance for pretrained models . The work is conceptually sound and original, with strong results, but I weighed it lower on clarity and practical impact due to inference cost and difficult presentation. The rebuttal partially addressed my concerns (efficiency clarification, sensitivity analysis, planned improvements), which improved my confidence, but did not fully resolve issues around clarity and usability. Overall, it reinforced my borderline assessment.

**Key Questions For Authors:**

- How sensitive is TIED to the Monte Carlo sample size N, diffusion schedule, and number of reverse steps in practice?
- Since results vary with the energy choice, can the authors better characterize when classifier-confidence energy is sufficient and when a learned prior such as VAE ELBO is necessary?

**Limitations:**

Not relevant here

**Strengths And Weaknesses:**

Strengths:
- Principled formulation of transformation inversion as posterior sampling on a Lie group, rather than direct reconstruction in data space.
- A reverse diffusion process that stays on-manifold and operates through Lie-algebra quantities.
- Good empirical results against several optimization and sampling baselines.

Weaknesses:
- Practical efficiency is a main concern: this is still an inference-time sampling method, so it is heavier than a standard forward pass or a lightweight canonicalization method.
- For a paper that studies image homographies, it is surprising that it does not show even one clear qualitative visual example of inverse homography recovery.
- I find the paper technically hard to understand, which may make it harder to reproduce and empirically validate.

---

> ### Author Rebuttal · Authors · 2026-03-31
>
> > W1. Efficiency: heavier than a forward pass or lightweight canonicalization?
>
> While overhead is unavoidable in any test-time method, including canonicalization, TIED remains efficient: time costs in Tab. 5, 6 show that TIED is comparable to the competing canonicalizations FoCal [1] and LieLAC [2], and slightly slower than ITS [3], which is specifically optimized for affine. TIED still outperforms these baselines in accuracy.
>
> If the reviewer has a specific lightweight canonicalization in mind, we are happy to discuss. Our canonicalization baselines are comprehensive, since any method that requires equivariant nets, e.g. [4-6], cannot be used for most of our tasks due to the challenging nature of the groups considered.
>
> Because our method is test-time, it does not require any retraining of the model, much like test-time methods for modern language models that are also heavier than a forward pass.
>
> > W2. Visual examples?
>
> We visualize affine/homography under classifier energy in [[PDF]](https://anonymous.4open.science/api/repo/tied-icml2026-94F8/file/tied_rebuttal_fig1_redacted.pdf?v=f3f5dfc5), showing that TIED sends data into in-distribution (corroborated by FIDs in Tab. 1 & 4, a perceptual metric).
>
> > W3. The paper is technically hard to understand (...) hard to reproduce and validate.
>
> We will improve the accessibility via a figure of diffusion on a Lie group, similar to [7-9] but adapted to trivialization (a draft: [[PDF]](https://anonymous.4open.science/api/repo/tied-icml2026-94F8/file/tied_rebuttal_fig2_redacted.pdf?v=7551ed04)). We will augment Alg. 1 with line-by-line descriptions [9], and improve proof sketches. While we cannot update the paper during the rebuttal, we are happy to elaborate on each point in the discussion phase. We will fully open-source the code.
>
> While our theory unavoidably uses some technical concepts, it is simpler than many prior works on manifold diffusion, e.g. [7] uses Girsanov theory, and [10] introduces an operator-theoretic generalization of scores to handle Lie groups. Our tools, including trivialization, are from standard differential geometry [11] and sufficiently handle arbitrary Lie groups.
>
> > Q1. Sensitivity?
>
> We study on affNIST under classifier energy (Sec. 5.2).
>
> **Table A.** Acc. (%) of TIED over N and M.
> |  | N=1 | N=2 | N=5 | N=10 |
> |---|---|---|---|---|
> | M=10 | 75.45 | 76.50 | 77.99 | 78.36 |
> | M=20 | 79.62 | 80.37 | 81.23 | 81.86 |
> | M=50 | 82.00 | 82.63 | 83.15 | 83.41 |
> | M=100 | 82.88 | 83.21 | 83.33 | 83.78 |
>
> For MC sample size N and sampling steps M, TIED scales smoothly with M, N. Even in the most resource-limited case (M=10, N=1), it achieves 75.45% accuracy and outperforms the baselines (ITS 67.18, FoCal 66.97, Langevin 53.98, LieLAC 73.79).
>
> **Table B.** Acc. (%) of TIED over noise schedules.
> | γmin, γmax | Acc. |
> |---|---|
> | 0.05, 1.0 | 83.24 |
> | 0.1, 1.0 | 82.63 |
> | 0.2, 1.0 | 81.45 |
> | 0.1, 0.5 | 79.74 |
> | 0.1, 2.0 | 77.97 |
> | 1.0, 1.0 | 60.00 |
> | 0.1, 0.1 | 59.80 |
>
> For diffusion schedule γ(t) = γmin (γmax/γmin)^t, setting γ(0) = γmin to be reasonably small, and γ(1) = γmax to be reasonably large, offers stable results, while enforcing γ(0) = γ(1) leads to degraded accuracy. This is expected from our variance-exploding formulation, which works the best when it anneals between large-diffusion noise and small perturbations of clean samples [12]. Enforcing it to only perform large diffusion does not allow diffusion to consolidate towards the clean samples, while only allowing small perturbations makes it inaccurate as the noising kernel no longer approximates the noise marginal well (l. 260-267).
>
> > Q2. When is classifier energy sufficient and when is a learned prior necessary?
>
> If one can afford a likelihood model, we consider using it beneficial, as it can induce a well-behaved score outside the data manifold and improve diffusion (as in our ELBO results). When these are not available, classifier energy, an implicit approx. of unnormalized likelihood [13], is still very useful to improve a pretrained classifier without additional training. TIED boosts the accuracy of a ResNet18 classifier from 55.48 to 82.64% on affNIST (Tab. 4), only using the classifier itself.
>
> ---
> [1] Singhal et al. Test-time canonicalization by foundation models, '25
>
> [2] Shumaylov et al. LieLAC, '25
>
> [3] Schmidt & Stober, Tilt your head, '24
>
> [4] Kaba et al. Equivariance with learned canonicalizations, '23
>
> [5] Kim et al. Learning probabilistic symmetrization, '23
>
> [6] Mondal et al. Equivariant adaptation of pretrained models, '23
>
> [7] De Bortoli et al. Riemannian score-generative modeling, '22
>
> [8] Chen & Lipman, Flow matching on geometries, '24
>
> [9] Zhu et al. Trivialized momentum facilitates diffusion on Lie groups, '25
>
> [10] Bertolini et al. Diffusion generative modeling on Lie group repr., '25
>
> [11] Lee, Introduction to smooth manifolds, '12
>
> [12] Song et al. Score-generative modeling through SDE, '21
>
> [13] Grathwohl et al. Your classifier is secretly an EBM, '20

---

> > ### Author Rebuttal · Reviewer_RyvQ · 2026-04-04
> >
> > Thank you for the detailed responses. The rebuttal addresses several of my concerns, though some reservations remain. The paper’s core formulation and application to test-time equivariance are clearly stated in the manuscript.
> >
> > W1. Efficiency
> > The clarification is helpful. However, I still think the paper should make the cost/accuracy tradeoff especially explicit, since practical adoption depends heavily on this.
> >
> > W2. Visual examples
> > The response is helpful, and it is good to know that qualitative affine/homography visualizations already exist in the supplement. I still believe at least one clear qualitative example should appear in the main paper, especially for homography inversion, since this would make the method much more interpretable to readers.
> >
> > W3. Accessibility / reproducibility
> > I appreciate the concrete plan to improve exposition with an added figure, more detailed algorithmic explanation, and clearer proof sketches. This would substantially improve the paper. My concern is reduced, though I still view presentation as a current weakness.
> >
> > Overall
> > The rebuttal improves my view of the paper, especially on empirical stability and on the role of different energies. My main remaining concern is presentation clarity, with a secondary concern around practical efficiency despite the additional runtime discussion. Overall, the rebuttal addresses several weaknesses, but I still view the current paper as borderline rather than clearly above threshold.

---

> > > ### Author Response · Authors · 2026-04-08
> > >
> > > We sincerely thank the reviewer for the thoughtful follow-up and careful consideration of the rebuttal. The feedback has been very helpful in improving the paper. We address these remaining points as follows.
> > >
> > > > W1. Efficiency The clarification is helpful. However, I still think the paper should make the cost/accuracy tradeoff especially explicit, since practical adoption depends heavily on this.
> > >
> > > We will explicitly include our responses to **W1** (efficiency) and **Q1** (cost-accuracy scaling) in Section 5.2 of the main paper. In these responses, we showed that (i) while overhead is unavoidable in any test-time method, TIED is efficient compared to competing canonicalization methods, and does not require retraining of the model, and that (ii) TIED scales smoothly with compute, while outperforming the baselines in the most resource-limited case on affNIST under classifier energy. We believe such efficiency makes TIED attractive as a practical test-time method for pretrained models for challenging symmetry groups found in computer vision and scientific applications.
> > >
> > > > W2. Visual examples The response is helpful, and it is good to know that qualitative affine/homography visualizations already exist in the supplement. I still believe at least one clear qualitative example should appear in the main paper, especially for homography inversion, since this would make the method much more interpretable to readers.
> > >
> > > We will move parts of the affine/homography visualizations in our response to original review's **W2** to Section 5.2 of the main text. We believe this will make our method substantially more interpretable.
> > >
> > > > W3. Accessibility / reproducibility I appreciate the concrete plan to improve exposition with an added figure, more detailed algorithmic explanation, and clearer proof sketches. This would substantially improve the paper. My concern is reduced, though I still view presentation as a current weakness.
> > >
> > > We will make sure to incorporate all the items in the updated version. We would like to use the rebuttal comment to line out the proof sketches more specifically.
> > >
> > > **Prop. 3.1 (Transformation inversion posterior).** For x̃ = g·x generated by an unknown g on an in-distribution data x, we want the posterior p(g|x̃) that inverts the data x̃ → g⁻¹·x̃ to be in-distribution. By writing down p(x,x̃|g) and marginalizing out x, we obtain that the likelihood p(x̃|g) is proportional to the data-space density p(g⁻¹·x̃) (up to a Jacobian correction). Then, Bayes’ rule gives the posterior p(g|x̃). Part 2 follows by changing variables from g to x' = g⁻¹·x̃, showing that canonicalized x’ land back in-distribution.
> > >
> > > **Prop. 3.2 (Equivariance of the posterior).** We want to show that transforming the input x̃ → h·x̃ shifts the posterior p(g|x̃) = p(hg|hx̃) accordingly, ensuring the canonicalization procedure is equivariant. By substituting x̃ → h·x̃ and g → hg into the posterior, the energy term is unchanged since (hg)⁻¹·(h·x̃) = g⁻¹·x̃. The extra Jacobian factor from h cancels between numerator and normalizer after a change of variable on the group and left-invariance of the Haar measure.
> > >
> > > **Prop. 4.1 (Reverse trivialized SDE).** Given the forward trivialized diffusion that noises group elements, we want to characterize its time-reversal for sampling. Since the forward process has no drift, its density evolves by the Brownian motion on the group. By checking that the proposed reverse drift (trivialized score) −γ²∇_𝔤 log p_t produces the same Kolmogorov forward equation (using that the divergence of p∇log p equals the Laplacian of p), the forward and reverse marginals are shown to match.
> > >
> > > **Prop. 4.2 (Trivialized target score identity).** We want to express the noisy score ∇_𝔤 log p_t in terms of the clean energy, without requiring clean samples. By writing p_t as a convolution of the clean density and the noise kernel, and taking the trivialized gradient inside the integral (which is valid because all gradients live in the same Lie algebra), we apply the log-derivative trick and Bayes' rule to obtain the score as a posterior-weighted average of clean energy gradients.
> > >
> > > **Prop. 4.3 (Monte Carlo score estimator).** We want a practical estimator of the score that only requires energy evaluations and noise samples. By substituting the Boltzmann form p₀ ∝ e⁻ᴱ into the convolution expression for p_t, the intractable normalizing constant cancels in the score. The remaining integral is an expectation over the noise kernel k_t, yielding a Monte Carlo estimator where the gradient sits outside a log-sum-exp over sampled noise terms.
> > >
> > > We believe these will significantly improve the clarity and accessibility of our results.

---

### Official Review · Reviewer_eNCP · 2026-03-12

**Soundness:** 3
**Presentation:** 3
**Significance:** 3
**Originality:** 4
**Overall Recommendation:** 4
**Confidence:** 2

**Summary:**

This study's core idea concerns recasting transformation inversion as posterior inference over Lie-group elements rather than reconstructing data directly in data space. This manuscript's specific area is diffusion-based sampling on Lie groups for test-time equivariance under unknown geometric or symmetry transformations. The paper models the posterior
p(g∣x~)
p(g∣
x
~
) with a Boltzmann density induced by a data-space energy, proves that sampling inverse transformations yields an equivariant randomized predictor, and introduces TIED, a reverse-time trivialized diffusion sampler that works on the group while computing scores in the Lie algebra. The main technical ingredients are a reverse trivialized SDE, a trivialized target-score identity, and a Monte Carlo estimator for the score. Empirically, the method is evaluated on synthetic sampling over
SO(10)
SO(10), affine and homography robustness for MNIST classifiers, and PDE symmetry tasks with neural operators, where it generally improves over optimization and sampling baselines.

**Compliance With Llm Reviewing Policy:**

Affirmed.

**Final Justification:**

No change after rebuttal.

**Key Questions For Authors:**

-

**Limitations:**

yes

**Strengths And Weaknesses:**

A major strength is that the paper has a coherent theoretical arc: it derives the transformation posterior on the group, proves an equivariance result for the induced predictor, and then introduces a diffusion sampler tailored to Lie groups via trivialization. The technical contribution appears substantive rather than incremental, especially the claim that the method handles general Lie groups without assuming compactness, a bi-invariant metric, or linear actions. The experiments are also reasonably broad for a theory-driven paper, spanning a synthetic multimodal sampling task, image transformations, and PDE symmetries. Originality is the strongest aspect of the paper. Modeling inversion as a posterior over transformations on the group, then sampling that posterior with a diffusion process that remains on-manifold and works through Lie-algebra computations, is an original and well-motivated combination. The reverse trivialized SDE, the trivialized target-score identity, and the Monte Carlo score estimator are, to my reading, the main novel pieces, and together they form a technically distinct contribution relative to deterministic canonicalization and prior Lie-group samplers.

---

> ### Author Rebuttal · Authors · 2026-03-31
>
> We are very glad that the reviewer has recognized our work as having a coherent motivation, a substantial technical contribution with strong originality, and a reasonably broad set of experiments. We are happy to address any remaining concerns or questions during the discussion period.

---

> > ### Author Rebuttal · Reviewer_eNCP · 2026-04-03
> >
> > To the best of my limited knowledge, the assigned score is appropriate.

---

> > > ### Author Response · Authors · 2026-04-08
> > >
> > > We sincerely thank the reviewer for the warm support and the thoughtful follow-up.

---

### Official Review · Reviewer_e3XS · 2026-03-16

**Soundness:** 3
**Presentation:** 3
**Significance:** 3
**Originality:** 3
**Overall Recommendation:** 5
**Confidence:** 3

**Summary:**

The paper tackles the general problem of transform inversion on Lie groups: given an observed sample that has been transformed by an unknown group element, the goal is to recover an inverse transform that maps it back to the data distribution.

The authors propose their method, TIED, to model a posterior over transformations. The method uses diffusion and the paper demonstrates improved canonicalization compared to optimization-based or deterministic baselines.

**Compliance With Llm Reviewing Policy:**

Affirmed.

**Final Justification:**

My concerns have largely been resolved. I am satisfied that the authors have adequately addressed the main questions regarding the contribution, and I recommend that they take a close look at the feedback and suggestions provided by the other reviewers when revising the submission.
I have updated my score and recommend this paper for acceptance.

**Key Questions For Authors:**

- The paper claims MCMC and even Langevin based methods are not possible. However, it seems on many groups we could do such sampling scheme, which could already provide a baseline, at least on Aff(2, R) and homography (perspective) transformations etc.

I am confused about the sentence:
 "the energy landscape can be highly rugged and multi-modal, especially when the energy is derived from a neural network"

Could authors elaborate on this a bit more? Firstly, MCMC-based/Langevin samplers should be able to sample from multi-modal distributions if they are not that hard (e.g. why would we expect distributions of simple tranformations on Aff(2,R) or homography transforms etc to be very rigged, right?), and also, I don't fully understand the "derived from a neural network" where I thought the attempt was to model the data-generating distribution. Or is the posterior dependent on the pre-trained model, in this case it would help if this point of the general set-up is made more clearly to avoid confusion.

**Limitations:**

-

**Strengths And Weaknesses:**

Strengths:
- The idea of running diffusion on Lie groups instead of data space is technically interesting.
- The method is general and can be applied in a wide range of settings.

Weaknesses:
- The intuition behind the diffusion process on the group manifold could be made more accessible.
- I didn't fully get why the "trivialized target-score" is called this way?
- Experimental section could use some more ablations (e.g. on noisy schedule, energy function) as well as simple comparison on MCMC-based approaches (see questions)
- In terms of baselines, missing standard MCMC-based baselines for sampling part and other works on equivariant canonicalization.

---

> ### Author Rebuttal · Authors · 2026-03-31
>
> > W1. Diffusion on groups could be more accessible.
>
> For accessibility, we will add a figure (a draft: [[PDF]](https://anonymous.4open.science/api/repo/tied-icml2026-94F8/file/tied_rebuttal_fig2_redacted.pdf?v=7551ed04)). We plan to show a Lie group, as a sphere, that transforms an image. On the sphere, we will show a distribution as a heatmap [1, 2], and mark a random group element as a point on the sphere. Then we will show the forward diffusion that starts with a sphere with a Boltzmann dist. and ends up with a sphere with noise. This would be a sequence of spheres with increasingly smoothed heatmaps and a group element moving along the stochastic trajectory. The image transformed by the group element would go from upright to randomly tilted.
>
> > W2. Why is trivialized target-score called this way?
>
> This is because the key object ∇𝔤 log p0 in Prop. 4.2 is the target score function, or the score of the target distribution p0, and it is trivialized, always taking values in the Lie algebra, not an arbitrary tangent space.
>
> The term “target score” is from [4] that expresses the noisy score ∇ log pt as an average of the clean (or, “target”) score ∇ log p0. When energy E = − log p0 is available, as in our setting, this is suitable. Yet, we want to sample from p0 on a Lie group G, which causes a problem: we can no longer average the target score over the group, as they live in different tangent spaces (l. 258-267 (right column)).
>
> The term “trivialization” is from a result in differential geometry that every tangent space of G can be one-to-one mapped to the Lie algebra g ≡ Rk, making them “trivial” [6]. This allows parameterizing any tangent vector as a vector in Rk, a method historically called trivialization [7, 8]. We use it to solve the issue of target scores. Prop. 4.2 shows that the trivialized noisy score ∇𝔤 log pt, in Rk, is an average of the trivialized target score ∇𝔤 log p0, which is also in Rk, and hence can be averaged across the group.
>
> > W3. Ablations?
>
> We add a study on affNIST under classifier energy (Sec. 5.2).
>
> **Table A.** Acc. (%) of TIED across noise schedules.
> | γmin, γmax | Acc. |
> |---|---|
> | 0.05, 1.0 | 83.24 |
> | 0.1, 1.0 | 82.63 |
> | 0.2, 1.0 | 81.45 |
> | 0.1, 0.5 | 79.74 |
> | 0.1, 2.0 | 77.97 |
> | 1.0, 1.0 | 60.00 |
> | 0.1, 0.1 | 59.80 |
>
> For our diffusion schedule γ(t) = γmin (γmax/γmin)^t, setting γ(0) = γmin to be reasonably small, and γ(1) = γmax to be reasonably large, offers stable results, while enforcing γ(0) = γ(1) leads to degraded accuracy. This is expected from our variance-exploding formulation, which works the best when it anneals between large-diffusion noise and small perturbations of clean samples [12]. Enforcing it to only perform large diffusion does not allow diffusion to consolidate towards the clean samples, while only allowing small perturbations makes it inaccurate as the noising kernel no longer approximates the marginal noise well (l.260-267).
>
> > W4/Q1. Baselines?
>
> **MCMC.** We do use an MCMC baseline, the kinetic Langevin, designed for Lie groups [7]. Our claim is that previous diffusion sampling could not be used (e.g. [3, 8]; l. 238-267 (right column)).
>
> **Other equivariant canonicalization.** Our baselines [7, 9-10] are comprehensive in terms of applicable methods, partially due to the challenging nature of the groups considered. Any method that requires an equivariant net [12-14], cannot be used for most of our experiments, as their constructions are not known.
>
> > Q2. Rugged energy / MCMC / setup?
>
> MCMC can indeed sample a multi-modal distribution if it is not too jagged. Fig. 3 uses kinetic Langevin MCMC [7] and confirms this. To clarify our setup, TIED (or other baselines) requires a data-space energy that quantifies unnormalized likelihood. For natural data, the true likelihood is not known, so one uses an approximation. If this is derived from a network, especially a classifier [16], this can indeed be very rugged, see Fig. 2. This poses a challenge to MCMC. Overall, TIED needs two components: an energy and a predictor. It’s possible to use the same model as the energy and predictor, e.g. a classifier.
>
> ---
> [1] De Bortoli et al. Riemannian score generative modeling, '22
>
> [2] Chen & Lipman, Flow matching on geometries, '24
>
> [3] De Bortoli et al. Target score matching, '24
>
> [4] Vincent, A connection between SM and DAEs, '11
>
> [5] Lee, Introduction to smooth manifolds, ’12
>
> [6] Lezcano-Casado, Trivializations for optimizations, '19
>
> [7] Kong & Tao, Kinetic Langevin MC, '24
>
> [8] Akhound-Sadegh et al. iDEM, '24
>
> [9] Schmidt & Stober, Tilt your head, '24
>
> [10] Singhal et al. Test-time canonicalization by FMs, '25
>
> [11] Shumaylov et al. LieLAC, '25
>
> [12] Kaba et al. Equivariance with learned canonicalization, '23
>
> [13] Kim et al. Learning probabilistic symmetrization, '23
>
> [14] Mondal et al. Equivariant adaptation of pretrained models, '23
>
> [15] Song et al. Score-based modeling through SDE, '21
>
> [16] Grathwohl et al. Your classifier is secretly an EBM, '20

---

> > ### Author Rebuttal · Reviewer_e3XS · 2026-04-07
> >
> > I thank the authors for their response.
> >
> > My concerns have largely been resolved. I am satisfied that the authors have adequately addressed the main questions regarding the contribution, and I recommend that they take a close look at the feedback and suggestions provided by the other reviewers when revising the submission. I have updated my score and recommend this paper for acceptance.

---

> > > ### Author Response · Authors · 2026-04-08
> > >
> > > We sincerely thank the reviewer for the warm support and the willingness to update the score. We will make sure to incorporate the feedback from reviews in our next revision.

---

### Official Review · Reviewer_3WQe · 2026-03-20

**Soundness:** 4
**Presentation:** 3
**Significance:** 3
**Originality:** 3
**Overall Recommendation:** 5
**Confidence:** 3

**Summary:**

This paper proposes TIED to address the problem of inverting an unknown Lie group transformation applied to a datum such that it is mapped back to the original data distribution, similar to canonicalization. This is useful for test-time equivariance, where pretrained models can be made robust to input transformations without retraining. The authors formulate the problem as sampling from a Bayesian posterior over Lie group transformations, where the posterior is a Boltzmann distribution defined by a data-space energy function. Assuming access to a differentiable energy function (e.g., from a pretrained classifier or VAE), they use a diffusion-based sampler that operates entirely on the Lie algebra via trivialization, avoiding the need to handle curved tangent spaces directly. A trivialized target score identity enables Monte Carlo estimation of the diffusion score without requiring training data from the posterior. Experiments on affine/homography transformations on MNIST and PDE solving with Lie point symmetries show that TIED consistently outperforms optimization and sampling baselines.

**Compliance With Llm Reviewing Policy:**

Affirmed.

**Final Justification:**

The rebuttal addressed all of my concerns and reinforced my assessment. The new ViT results on more realistic datasets are compelling.

**Key Questions For Authors:**

1. Proposition 3.1 seems to suggest that the posterior isn't well defined on groups with nontrivial stabilizers (e.g. the digit "8" with 180 degree rotations). Can you modify TIED so that the posterior is still well-defined (maybe use the quotient group G/Stab(x)), and would TIED be able to sample from all the the modes?

**Limitations:**

The authors adequately discuss the limitations of their method.

**Strengths And Weaknesses:**

Strengths:
- Clear problem formulation and motivation (on test-time equivariance). Framing transformation inversion as posterior inference with an energy-based model is elegant and well-motivated.
- The trivialized target score identity sidesteps the issue of parallel transport and doesn't require bi-invariant metric. Even works for noncompact Lie groups. I think this is possibly the strongest feature of this method.
- Specifically on MNIST with affine transforms, TIED outperforms significantly. TIED also seems to have comparable wall clock times to other baselines.

Weaknesses:
- While the authors mention that "the energy only needs to be accurate within each orbit", the method's success seems to hinge on the accuracy of the energy function. Comparing VAE ELBO vs classifier logits on MNIST, the VAE ELBO seems to perform much better. I wonder how sensitive the method is to energy mispecification and it would be nice to validate the "accuracy within each orbit" statement.
- The experiments consider two different settings: transformed MNIST and Lie point symmetries, but I'd be interested in a more realistic image dataset (e.g. CIFAR10) or harder PDEs. Also, since the paper talks about pretrained models, perhaps a larger transformer model instead of ResNet18 would have been better. Not a big weakness though.
- The method needs O(M*N) energy evaluations per test sample. An accuracy plot for varying M and N could be useful.

---

> ### Author Rebuttal · Authors · 2026-03-31
>
> > W1.Sensitivity to energy misspecification? Validity of "accuracy within each orbit" statement?
>
> Our statement that the energy only needs to be accurate within each orbit is a mathematical one and can be proven to be valid (we leave a proof below). Hence, choosing a better energy in our framework means choosing a more accurate energy within each orbit. A misspecified energy would result in sampling from the posterior for a misspecified likelihood, affecting the transformations. Compared to VAE ELBO energy, classifier energy seems to be a low-quality surrogate because the classifier has never seen inputs outside of training datapoints, and the energies are solely based on positive or canonical samples. Nevertheless, we remark that all considered test-time baselines (ITS [1], FoCal [2], kinetic Langevin [3], and LieLAC [4]) depend on the accuracy of the energy in the same manner. Furthermore, our method outperforms them under both high-quality and low-quality energies.
>
> **Proposition.** Consider two energy functions U, U’: X -> R that are equal up to additive constant only on each orbit, and not across different orbits. They induce the same Boltzmann density p(g|x).
>
> Proof. This follows directly from our formula in l. 228-229 that the induced energy on the group is given as $E(g) = U(g^{-1}x) − \log |\det J_{g^{-1}}(x)|$, combined with the assumption that $U(gx) = U’(gx) + C$ for a fixed $x$ and any $g$, yielding $E’(g) = U'(g^{-1}x) − \log |\det J_{g^{-1}}(x)| = E(g) - C$.
>
> > W2. The experiments consider two different settings: transformed MNIST and Lie point symmetries, but I'd be interested in a more realistic image dataset (e.g. CIFAR10) or harder PDEs. Also, since the paper talks about pretrained models, perhaps a larger transformer model instead of ResNet18 would have been better. Not a big weakness though.
>
> We are currently carrying out some of the requested experiments, and will report during the discussion period once we have the results.
>
> > W3. An accuracy plot for varying M and N?
>
> We add a study on affNIST under classifier energy (Sec. 5.2). TIED scales smoothly with M and N. Even in the most resource-limited regime (M=10, N=1), it achieves 75.45% accuracy and outperforms the baselines (ITS 67.18, FoCal 66.97, kinetic Langevin 53.98, LieLAC 73.79).
>
> **Table A.** Acc. (%) of TIED across MC sample size (N) and # sampling steps (M).
> |  | N=1 | N=2 | N=5 | N=10 |
> |---|---|---|---|---|
> | M=10 | 75.45 | 76.50 | 77.99 | 78.36 |
> | M=20 | 79.62 | 80.37 | 81.23 | 81.86 |
> | M=50 | 82.00 | 82.63 | 83.15 | 83.41 |
> | M=100 | 82.88 | 83.21 | 83.33 | 83.78 |
>
> > Q1. Prop. 3.1 seems to suggest that the posterior isn't well defined on groups with nontrivial stabilizers (e.g. the digit "8" with 180 degree rotations). Can you modify TIED so that the posterior is still well-defined (maybe use the quotient group G/Stab(x)), and would TIED be able to sample from all the modes?
>
> We thank the reviewer for this insightful comment and suggestion. Unfortunately the quotient $G/Stab(x)$ is only defined as a group when $Stab(x)$ is normal which is not always guaranteed.
>
> The freeness assumption is however mainly a technical condition used for the proof of the theorem. In practice, our method does not require freeness, as it only relies on defining and sampling an energy over $G$.
> When the action is not free, the main effect is redundancy: multiple group elements correspond to the same transformed sample. This does not affect recovery of $g^{-1}\cdot \tilde x$, which is invariant along stabilizer directions. In other words, not sampling from all the modes is not a problem because the different modes are associated with identical copies of the data.
>
> The only potential issue we see is that noncompact stabilizers can lead to a non-normalizable posterior over G. In those cases the density is not well-defined. However, even in this situation, the induced sample $g^{-1}\cdot \tilde x$ remains unchanged along those directions, so this does not impact recovery in practice.
>
> Finally, in the applications we consider, the freeness assumption holds. It is violated when the input data has perfect symmetry, which would be artificial.
>
> Overall, the freeness assumption simplifies the theory, but is not a limitation in practice.
>
> ---
> [1] Schmidt & Stober, Tilt your head, '24
>
> [2] Singhal et al. Test-time canonicalization by foundation models for robust perception, '25
>
> [3] Kong & Tao, Convergence of kinetic Langevin MC on Lie groups, '24
>
> [4] Shumaylov et al. LieLAC, '25

---

> > ### Author Rebuttal · Reviewer_3WQe · 2026-04-03
> >
> > I appreciate the author's thoughtful and clear response. My concerns are mostly resolved. I maintain my score of Accept. As for why not raising my score, I don't think this paper achieves the bar of "flawless paper, exceptional impact in one or more areas of AI" due to its relatively limited applicability (symmetrizing pretrained models).

---

> > > ### Author Response · Authors · 2026-04-08
> > >
> > > We sincerely thank the reviewer for the thoughtful follow-up and careful consideration of the rebuttal. We will make sure to incorporate the content of the rebuttal to the next revision.
> > >
> > > Regarding W2 (a more realistic image dataset and a larger transformer model), we ran an additional experiment using a vision transformer (ViT) model on the affine-transformed Imagenette 320 dataset* with realistic natural images in ten classes. We took a 21M-parameter ViT pretrained on ImageNet-22k [1] and fine-tuned it as a classifier on the clean training images resized to 224x224 until it reached 99% accuracy. Then we evaluated the classifier on affine-transformed images from the test set (we used a fixed random subset of 100 images due to cost constraints). For all transformation inversion methods, we used the classifier confidence energy. The results are below, and largely agree with our findings in Section 5.2.
> > >
> > > **Table B.** Imagenette 320 classification test accuracy.
> > > |  | Accuracy |
> > > |---|---|
> > > | ViT | 80% |
> > > | ViT + ITS | 39% |
> > > | ViT + FoCal | 91% |
> > > | ViT + Kinetic Langevin | 89% |
> > > | ViT + LieLAC | 97% |
> > > | ViT + TIED (Ours) | **98%** |
> > >
> > > **Table C.** Imagenette 320 classification wall-clock time.
> > > |  | Wall-clock time |
> > > |---|---|
> > > | ITS | 5m 15s |
> > > | FoCal | 3m 3s |
> > > | Kinetic Langevin | 1h 24m 2s |
> > > | LieLAC | 23m 25s |
> > > | TIED (Ours) | 23m 31s |
> > >
> > > *We initially also considered CIFAR10, but found it problematic since its low resolution (32x32) caused destructive and irreversible loss of visual details under affine transformations and pixel resampling.
> > >
> > > [1] timm/tiny_vit_21m_224.dist_in22k_ft_in1k

---

### Official Review · Reviewer_tryP · 2026-03-22

**Soundness:** 3
**Presentation:** 4
**Significance:** 3
**Originality:** 3
**Overall Recommendation:** 5
**Confidence:** 4

**Summary:**

This paper introduces a new method to increase equivariance with respect to group transformation of a pretrained model at test time. The suggested method is to infer a group transformation which canonicalizes (transforms back into the training distribution) a transformed input sample and then apply the pretrained model on the back-transformed, in-distribution sample. Since inferring this group transformation constitutes a blind inverse problem, the authors suggest to use a Bayesian probabilistic approach and sample the group element from the posterior over the symmetry group. The posterior is modeled as a Boltzmann distribution. The main technical contribution of the paper is a diffusion model on the Lie algebra of the symmetry group to generate samples from this distribution given the energy function of the prior ove the data. The diffusion model is trained with score matching in the Lie algebra, using the trivialization map of the group.

**Compliance With Llm Reviewing Policy:**

Affirmed.

**Final Justification:**

The rebuttal answered my questions and increased my confidence in my assessment of the paper. This is a meaningful contribution to a relevant field of machine learning. The presentation is excellent, soundness, originality and significance are good. However, the paper addresses only a subfield of machine learning, and the impact is not exceptional. Therefore, I retain my assessment of accept but not strong accept.

**Key Questions For Authors:**

1. In l. 218 (left column) you mention averaging over several sampled group elements to improve performance. Did you do this in your experiments? If so, how many samples did you use?
1. How compute-intensive is your algorithm, compared to the other algorithms in the experiments? Since the canonicalization algorithms you studied use considerable test-time compute, it would be interesting to see the differences between the methods. Also, letting optimization algorithms run longer or using more samples for the sampling-based algorithms could improve performance, making compute constraints an important part of the overall evaluation. A comparison to the manifestly equivariant models in the MNIST experiment in this regard would be interesting as well since these architectures are known to be more compute intensive but do not involve optimization or sampling at test time. These trade-offs affect the significance of the proposed method.
1. How significant is the limitation that the group needs to act freely on the data space (Proposition 3.1)? Are there important application cases where this is not the case?

**Typos**
- Caption Fig. 2: Dot missing after "transformations"
- l. 343 (left column) remove "a"
- l. 367 (left column) "and" missing after "images"

**Limitations:**

yes

**Strengths And Weaknesses:**

**Soundness**
All claims are well supported. Extensive appendices contains proofs of all statements, although I did not check them in detail. Experimental results include ablations over several baselines covering a variety of approaches to solve the problem. The compute efficiency of the method is not discussed and is an important aspect of the overall evaluation (see my question below).

**Presentation**
The presentation is excellent. The main text is clearly structured and focuses on the main points of the method. The necessary background is provided and for technical results, intuitive explanations are given. The only information I was missing is which groups do not admit a bi-invariant metric (since the proposed diffusion model seems to be go beyond other models available in the literature only in those cases).

**Significance**
How symmetries are best incorporated into deep learning models is an important and active research field with many application domains. This paper suggest a novel technique in this field. However, the exact tradeoffs of this method versus other methods are a bit unclear to me, see my question below.

**Originality**
I am no expert in the field of canonicalization or diffusion models on Lie groups, but the main novelty seems to be two-fold:
- Using a diffusion model to sample from the posterior over group elements
- The introduction of a diffusion model on Lie algebras which relies on the trivialization map. Using the trivialization maps has some technical advantages (it does not require a bi-invariant metric).

---

> ### Author Rebuttal · Authors · 2026-03-31
>
> > W1. Bi-invariant metric?
>
> A Lie group G admits a bi-invariant metric iff G ≡  Rk × K, with K compact ([1], Lemma 7.5). This is a strong constraint. Many practical groups fail this, including the Euclidean group, the general linear group in d > 1, the affine group, and the projective group (used in our experiments). Many Lie point symmetries of PDEs, relevant to scientific applications, also fall outside.
>
> > Q1. Averaging?
>
> We indeed tried MC averaging and found it improves performance. We also found that an alternative ensemble scheme is to, after diffusion sampling, select the sample with lowest energy. This has a slightly lower cost compared to averaging and performs better (70.16 -> 82.64% acc., affNIST with classifier energy). This can be interpreted as sampling from a sharpened distribution (CDF raised to a power) then doing one-sample MC. It preserves equivariance (proof below, to be included in the appendix). This is the scheme we use to obtain our final results and will be clarified in the main text. We use N = 64, matching LieLAC’s multiple initializations ([2], p.8). We also take this cost of ensembling into account when comparing to other methods.
>
> **Proposition.** Given a G-equivariant p(g|x), define p(g*|x) by sampling g1, …, gN independently conditioned on x and taking the unique (a.s.) max-likelihood g*. Then p(g*|x) is G-equivariant.
>
> Proof. By equivariance, $g\_i|h\cdot x =d h\cdot\tilde{g}\_i$ with $\tilde{g}\_i\sim p(\cdot|x)$ i.i.d., so $g^* | h\cdot x =d \arg\max\_i\, p(h\cdot\tilde{g}\_i|h\cdot x) = h\cdot\arg\max\_i\, p(\tilde{g}\_i|x) = h\cdot(g^*|x)$.
>
> > Q2a. Efficiency? Compared to others?
>
> As in Tab. 5 & 6, TIED’s time cost is lower than FoCal [4] and kinetic Langevin [5] (thanks to parallelization) and comparable to LieLAC [2], while outperforming all in accuracy. TIED is a bit more costly than ITS [3], though we remark that ITS [3] is specialized for affine inversions. We use 8 search levels for ITS, which we found necessary for a reasonable result, a more generous resource allocation than the original paper [3] which considered 3-5 levels. Increasing the search levels or other parameters did not lead to further gains, consistent with [3], Fig. 11.
>
> While we restrict the cost of our method to be comparable with baselines, we do find that its performance scales smoothly with compute. To illustrate this, we perform a study on affNIST with classifier energy (Sec. 5.2), varying the MC sample size for score estimation (N) and the number of sampling steps (M). Even in most resource-limited regime (M=10, N=1), it achieves 75.45% accuracy and outperforms the baselines (ITS 67.18, FoCal 66.97, kinetic Langevin 53.98, LieLAC 73.79).
>
> **Table A.** Acc. (%) of TIED across MC sample size (N) and sampling steps (M).
> |  | N=1 | N=2 | N=5 | N=10 |
> |---|---|---|---|---|
> | M=10 | 75.45 | 76.50 | 77.99 | 78.36 |
> | M=20 | 79.62 | 80.37 | 81.23 | 81.86 |
> | M=50 | 82.00 | 82.63 | 83.15 | 83.41 |
> | M=100 | 82.88 | 83.21 | 83.33 | 83.78 |
>
> > Q2b. Equivariant models in MNIST and tradeoffs?
>
> The MNIST experiment compares with equivariant models aff/homConv [6], specialized for affine and projective groups. While these do not perform test-time optimization, their training is costly, taking 36 hours each (Tab. 5). Our method is thus most practical when training an equivariant network is challenging, either due to cost (e.g. aff/homNIST) or a lack of equivariant architecture (e.g. PDE symmetries), and a pretrained/foundation model is available. Among the considered test-time methods, TIED is computationally efficient (Tab. 5).
>
> > Q3. Free-action assumption (Prop. 3.1)?
>
> The assumption is mainly a technical condition used for the proof. In practice, TIED does not require freeness, as it only relies on defining and sampling an energy over G. When the action is not free, the main effect is redundancy: multiple group elements correspond to the same transformed sample. This does not affect recovery of $g^{-1}\cdot \tilde x$, which is invariant along stabilizer directions.
>
> The only potential issue we see is that noncompact stabilizers can lead to a non-normalizable posterior over G. In those cases the density is not well-defined mathematically (but the sampling scheme is still well-defined). Even in this situation, the induced sample $g^{-1}\cdot \tilde x$ remains unchanged along those directions, so this does not impact recovery in practice.
>
> Finally, in the applications we consider, the freeness assumption holds. It is violated when the input data has perfect symmetry, which would be artificial. Overall, the freeness assumption simplifies the theory, but is not a limitation in practice.
>
> ---
> [1] Milnor, Curvatures of left invariant metrics on Lie groups, ‘76
>
> [2] Shumaylov et al. LieLAC, '25
>
> [3] Schmidt & Stober, Tilt your head, '24
>
> [4] Singhal et al. Test-time canonicalization by FMs, '25
>
> [5] Kong & Tao, Kinetic Langevin MC, '24
>
> [6] MacDonald et al. Enabling equivariance for arbitrary Lie groups, '22

---

> > ### Author Rebuttal · Reviewer_tryP · 2026-04-01
> >
> > Thank you for thoroughly answering my questions. Please provide a summary in the camera ready version of the paper.
> > I will leave my score at the accept level, since I do not think that the criterion of "exceptional impact on one or more areas of AI"  for strong accept is satisfied here. I will however increase my confidence score.

---

> > > ### Author Response · Authors · 2026-04-08
> > >
> > > We sincerely thank the reviewer for the willingness to update the confidence score. We will make sure to incorporate the content of the rebuttal to the next (camera ready) revision.

---

### Decision · Program_Chairs · 2026-04-30

**Decision:**

Accept (regular)

**Comment:**

The paper formulates transformation inversion as Bayesian inference over Lie group elements: given a transformed input, the goal is to infer an inverse transformation that maps it back to the data distribution. The posterior over transformations is defined as a Boltzmann distribution induced by a data-space energy function. To sample from this posterior, the authors introduce a diffusion-based sampler on Lie groups that operates in the Lie algebra via trivialization. A key contribution is a target-score identity that enables reverse diffusion using only the energy function, without requiring samples from the posterior. This framework improves test-time equivariance of pretrained models, with empirical gains on image classification under homographies and PDE symmetry tasks.

The proposed idea is elegant, technically sound, and supported by solid theoretical results. The paper studies an interesting and timely problem, and the motivation is clearly articulated. A notable strength is the introduction of a novel perspective that models a posterior over transformations through a diffusion-based approach. In particular, the development of a diffusion sampler on Lie groups via Lie algebra trivialization constitutes a significant technical contribution. The experimental validation is thorough and well designed, supporting the claims made in the paper. Overall, the paper is well written, although its accessibility could be further improved for a broader audience.

The paper would benefit from a clearer discussion of the trade-offs compared to existing approaches. As an inference-time sampling method, the proposed approach is inherently more computationally demanding than a standard forward pass or lightweight canonicalization techniques. Additionally, it relies on access to an accurate energy-based model, which could limit its applicability. From an empirical perspective, the evaluation could be strengthened by including more challenging tasks, additional ablation studies, and broader comparisons to existing methods.

The concerns raised by the reviewers have been addressed to a significant extent during the rebuttal phase, which is appreciated. The overall consensus is that this paper constitutes a significant contribution to a timely problem, proposing an elegant solution supported by both theoretical and empirical results. The main remaining concern is the accessibility of the paper, which should be improved. For these reasons, I recommend acceptance, and I encourage the authors to incorporate the reviewers’ feedback and improve the paper accordingly.